# Human and macaque pairs employ different coordination strategies in a transparent decision game

**Sebastian Moeller[1,2], Anton M Unakafov[1,2,3,4,5], Julia Fischer[2,6,7], Alexander Gail[1,2,3,8†], Stefan Treue[1,2,3,8†], Igor Kagan[1,2*†]**

[1]Cognitive Neuroscience Laboratory, German Primate Center – Leibniz Institute for Primate Research, Göttingen, Germany; [2]Leibniz ScienceCampus Primate Cognition, Göttingen, Germany; [3]Georg-Elias-Müller-Institute of Psychology, University of Gottingen, Göttingen, Germany; [4]Max Planck Institute for Dynamics and Self-Organization, Göttingen, Germany; [5]Campus Institute for Dynamics of Biological Networks, Gottingen, Germany; [6]Cognitive Ethology Laboratory, German Primate Center – Leibniz Institute for Primate Research, Göttingen, Germany; [7]Department of Primate Cognition, Johann-Friedrich-Blumenbach Institute for Zoology and Anthropology, University of Gottingen, Göttingen, Germany; [8]Bernstein Center for Computational Neuroscience, Göttingen, Germany

**Abstract** Many real-world decisions in social contexts are made while observing a partner's actions. To study dynamic interactions during such decisions, we developed a setup where two agents seated face-to-face to engage in game-theoretical tasks on a shared transparent touchscreen display ('transparent games'). We compared human and macaque pairs in a transparent version of the coordination game 'Bach-or-Stravinsky', which entails a conflict about which of two individually-preferred opposing options to choose to achieve coordination. Most human pairs developed coordinated behavior and adopted dynamic turn-taking to equalize the payoffs. All macaque pairs converged on simpler, static coordination. Remarkably, two animals learned to coordinate dynamically after training with a human confederate. This pair selected the faster agent's preferred option, exhibiting turn-taking behavior that was captured by modeling the visibility of the partner's action before one's own movement. Such competitive turn-taking was unlike the prosocial turn-taking in humans, who equally often initiated switches to and from their preferred option. Thus, the dynamic coordination is not restricted to humans but can occur on the background of different social attitudes and cognitive capacities in rhesus monkeys. Overall, our results illustrate how action visibility promotes the emergence and maintenance of coordination when agents can observe and time their mutual actions.

**\*For correspondence:**
ikagan@dpz.eu

†Co-senior authors

**Competing interest:** The authors declare that no competing interests exist.

## Editor's evaluation

This study investigates and compares spontaneous turn-taking behavior in pairs of macaque monkeys and human participants. The study is well-designed and uses a novel format for dynamic interaction. The analyses are rigorous and support the overall conclusion that there are differences between species in their tendencies toward cooperative, mutually beneficial behaviors, with humans exhibiting more prosocial tendencies. This finding, as well as the rich description of pair interactions in each species, is likely to be relevant to a broad range of researchers interested in social behavior.

**eLife digest** To live with others is to make concessions. You may want to go to the movies tonight, but your partner may prefer the theatre: reaching a mutually desirable goal – that is, spending time together – requires adjusting your preferences to theirs. Many other social species also make such decisions, in particular monkeys that live in large groups.

Conceptually, these interactions are known as coordination games. In such scenarios, two players must coordinate their actions to attain a coveted reward, but they must also resolve a conflict about who gets the larger share. This makes the joint strategy non-trivial, and different pairs of players might resort to different strategies. In the laboratory, coordination games are often tested in settings which do not allow participants to monitor each other's behaviors as they make these complex choices. In real life, however, individuals making a joint decision can often observe each other and receive immediate feedback.

In response, Moeller et al. developed a new way to test coordination games that allows more realistic social interactions. In their setup, two participants face each other and use a shared see-through touchscreen to perform a task. This new design was used to test how humans and macaque monkeys solved a simplified version of the 'Bach or Stravinsky' coordination game, which involves choosing between a red and blue target on the screen. Players in a pair had been trained to 'prefer' opposite colors. In this game, collaboration is beneficial (both individuals get a better prize if they choose the same color) but also creates unfairness (the reward is higher for the participant whose 'favorite' color is selected).

When paired up, both humans and monkeys learned to collaborate and to go for the same color (or, in some monkey pairs, the same side of the screen). However, only humans took turns selecting red or blue so that players could alternate getting the highest reward. Monkeys usually settled on one color throughout the game, unless they had learned the 'turn-taking' strategy from a human partner; in that case, the color chosen in each trial was typically determined by the monkey who was the faster to move.

These experiments show how monkeys and humans use visual information about their partner's actions to coordinate their choices, paving the way for further decision-making studies that accurately reflect how interactions unfold in real life. Moeller et al. expect that this will help to understand how cooperation and competition emerge in these two species, including how direct face-to-face contact, or lack thereof in some aspects of our modern world, shapes our social behavior.

## Introduction

The majority of primate species live in complex social groups, in which interactions range from intense competition to cooperation (*Byrne and Bates, 2010*; *Chang et al., 2013*; *Jaeggi et al., 2010*; *Joly et al., 2017*; *Platt et al., 2016*). To adjust behavior optimally, individuals need to assess their own actions and goals, and those of other group members, while taking into account the history of interactions with and between these individuals (*Frith and Frith, 2010*; *Platt et al., 2016*; *de Waal, 2007*). Coordination is essential for maintaining cohesion between group members, avoiding conflicts, and achieving individual and joint action goals. Here, we focus on understanding how such coordination can be achieved and maintained in a dyadic 'transparent' setting in which agents can observe each other's ongoing actions and social cues (*Unakafov et al., 2020*; *Unakafov et al., 2019*), like in many natural situations (*Conty et al., 2012*; *Fruteau et al., 2013*; *Rizzolatti et al., 2001*; *Silk, 2007*).

Game theory, developed to study strategic interactions in rational decision-makers (*Von Neumann and Morgenstern, 1944*), offers a powerful framework to investigate dyadic social interactions (*Rilling and Sanfey, 2011*; *Sanfey, 2007*; *Tremblay et al., 2017*) and the evolution of coordination (*Brosnan, 2018*; *Santos and Rosati, 2015*; *Smith, 1979*). In 2×2 games (*Rapoport et al., 1976*; *Rapoport, 1973*), each of the two agents chooses one of two actions and the outcome depends on the combination of their choices. Non-zero-sum 2×2 games that have both cooperative and competitive elements with aligned and opposing interests are particularly interesting for understanding many realistic scenarios. Here, the agents need to coordinate their choices to maximize individual and/or joint rewards (*Rapoport, 1967*). For example, in the Stag Hunt (or Assurance) game, agents choose between maximizing the individual as well as joint reward and minimizing the individual risk

(*Skyrms, 2003*). Humans (*Homo sapiens*), chimpanzees (*Pan troglodytes*), rhesus macaques (*Macaca mulatta*), and capuchin monkeys (*Cebus apella*) coordinate at the above chance level on mutually beneficial high-reward/high-risk choices in the iterated Stag Hunt (*Brosnan et al., 2012*; *Brosnan et al., 2011*; *Bullinger et al., 2011*; *Duguid et al., 2014*). Similar to Stag Hunt, in Prisoner's Dilemma (PD) two agents maximize their *joint* reward if they both choose to cooperate. Yet, an individual agent obtains the highest reward when defecting while the partner cooperates (and receives the lowest reward), creating a dilemma between the selfish temptation to defect and a riskier cooperation that requires trust. Humans often show a bias towards cooperative behavior (*Camerer, 2003*; *Rilling et al., 2002*; *Wedekind and Milinski, 1996*). Remarkably, it was demonstrated that macaques, while mostly defecting, chose mutual cooperation significantly more and mutual defection significantly less than expected by chance, especially after preceding cooperation, in an iterated PD game (*Haroush and Williams, 2015*). This finding suggests that macaques can at least partially overcome the selfish motives and reciprocate cooperation.

Another type of conflict that contrasts selfish and prosocial preferences is implemented in the Conflict game (also known as Hawk-Dove, or game of Chicken) (*Brosnan et al., 2017*; *Smith, 1979*). These anti-coordination games model a competition over a shared resource that can be monopolized by one agent, while an actual clash is highly detrimental to both. Humans, capuchins, and macaques often converge on anti-coordination (*Brosnan et al., 2017*), but importantly, only humans alternate between the two individually-optimal anti-coordinated choices (*Brosnan et al., 2017*; *Grueneisen and Tomasello, 2017*; *Helbing et al., 2005*). Such alternating behavior results in the maximal and equal payoff for both agents and is called 'cooperative turn-taking' (*Bornstein et al., 1997*; *Colman and Browning, 2009*; *Noë, 2006*).

In the above games, reward-optimizing rational agents should coordinate on one option (Stag Hunt and Prisoner's Dilemma) or anti-coordinate on opposite options (Conflict game). A third, less studied type of game that emphasizes the coordination of *either of two options* is known as Battle of the Sexes, or Bach-or-Stravinsky game (BoS) (*Kilgour and Fraser, 1988*). Each agent has an individually preferred option, but coordinating on either one of these options adds the same bonus to both agents. This renders any coordinated choice better than no coordination for both agents, but one coordinated choice is better for the first agent and the other better for the second agent. The rational choice is to coordinate, but unlike the conceptually simpler optimal convergence on one option in Stag Hunt and PD, BoS includes an inherent conflict about who profits the most. This combination of cooperation and conflict offers an interesting opportunity for studying social interactions in nonhuman primates. As in the Conflict game, games of the BoS type revealed that humans often take turns, switching between the two coordinated options to ensure fairness (*Ioannou and Romero, 2014*; *Sonsino and Sirota, 2003*). Interestingly, in a rope-pulling task that requires cooperation but rewards only one agent at a time, trial-by-trial turn-taking frequently took place in 5-year-old children, but neither in 3-year-old children nor in chimpanzees (*Melis et al., 2016*). This result begs the question whether turn-taking requires special social or cognitive abilities that are unique to humans. We therefore compared the behavior of (adult) humans and rhesus macaques in the BoS economic game.

Traditionally, economic games are played either '*simultaneously*' (neither agent knows the choice of the other before making its own decision) or *sequentially* in a predetermined order. Yet, real dyadic interactions often play out with the partner's actions in direct sight (*Dugatkin et al., 1992*; *van Doorn et al., 2014*). Thus, the timing of one's own and the other's actions becomes part of the strategy space (*McDonald et al., 2019*). Such a 'transparent' continuous time setting can change choice strategies as compared to the classic simultaneous and sequential settings (*Noë, 2006*).

To account for changes in strategies during action visibility compared to 'opaque' (simultaneous) choices, we recently developed the concept of 'transparent games' that extends the classical evolutionary game-theoretic analysis to real-time interactions in which the visibility of partners' actions depends on their relative action times (*Unakafov et al., 2020*; *Unakafov et al., 2019*). Here, we use the theoretical insights from that work and analyze short- and long-term dynamics of choices, mutual information, and action times in humans and rhesus monkeys playing a transparent version of an iterated movement-based BoS game. To this end, we designed a novel dyadic interaction platform where two human or monkey agents face each other and act on the same visual objects on a vertical touch-sensitive transparent display between them. This configuration provides action visibility, i.e. it allows

agents to monitor and react to each other's actions in real-time, emulating naturalistic interactions under well-controlled laboratory conditions.

To facilitate inter-species comparability, we paired human or macaque agents without explicit task instructions. We expected that the transparency and instantaneous coaction (*van Doorn et al., 2014*) would facilitate efficient coordination in both species. Based on their cognitive abilities (*Proto et al., 2019*), propensity for fairness (*Peysakhovich et al., 2014*), and perspective-taking (*Devaine et al., 2017*; *Herrmann et al., 2007*), we predicted humans to engage in pro-social turn-taking, where the reward maximization is alternated between partners. We predicted that macaques would also utilize the information about the partner's actions, but would be less likely to grant an advantage to the other player. Instead, we expected that the monkeys would coordinate to increase individual (and joint) rewards, but without turn-taking to balance rewards between players.

## Results

Two human or macaque agents sat face-to-face and performed a task on a dyadic interaction platform using a shared transparent vertical workspace (*Figure 1A*). Agents could simultaneously see the stimuli and each other's actions. In our variant of the Bach-or-Stravinsky (BoS) paradigm (Materials and methods), each agent could choose one of two simultaneously presented color targets (red and blue, *Figure 1B*). The left/right position of each color was randomized across trials (50% left, 50% right). The rewards followed the payoff matrix shown in *Figure 1C* and depended on the choices of both agents. Prior to the first dyadic session, each human or macaque agent in a pair was individually trained to associate one of the two color targets with a larger reward, so that the two agents preferred different colors. In the dyadic trials, selecting the same target resulted in an additional reward on top of the individually trained values - such that choosing the same target was always better than choosing different targets, for both agents individually, and as a group. But in such coordinated trials, the agent whose individually preferred ('own') color was selected receives a larger reward. Therefore, agents may attempt to behave selfishly/competitively in order to gain more by insisting on their own target color – which pays off as long as the partner accommodates. This paradigm probes the ability to coordinate in order to obtain higher rewards, and the capacity to deal with the unfairness introduced in each coordinated trial.

### Human pairs mainly converge on fair coordination

We recruited 38 naïve human subjects to assess performance in the transparent BoS paradigm (*Supplementary file 1*). *Figure 1E and F* illustrates an exemplary human pair that developed such alternating turn-taking behavior, showing the *fraction of choosing own* color (FCO) - the likelihood for each agent to select their own preferred color, and the *fraction of choosing left* side (FCL) - the likelihood for each agent to select the target on the left side of the display. These measures provide direct insight into each agent's target and side biases. Here, the FCO curves show that this pair mostly jointly selected A's or B's color. After an initial period of long blocks of trials, the pair switched to shorter blocks. The FCL curves show that the agents did not have a side bias, as was also the case for the majority of pairs (*Figure 2—figure supplement 1B*).

Human dyads showed mainly one of three different choice patterns. *Figure 2A* plots the larger of the two average fractions of choosing own values in a pair versus the smaller of the two, for all 19 pairs. The FCOs of 53% (10 of 19 pairs, teal underlay) had balanced values close to 0.5 for both agents, indicating alternating between colors, using either trial-by-trial or block-wise turn-taking. The remaining 47% (9 of 19 pairs) had FCO of at least one agent close to one or to zero (both cases indicate a fixed color selection). Five pairs coordinated on a fixed color favoring one agent (blue underlay). Of the four pairs that did not coordinate successfully, in two pairs both agents selected only their own preferred color (red underlay). In the remaining two pairs, one agent selected the non-preferred color while the other switched between the two colors or selected one side. Based on the debriefing responses (Appendix 1, *Supplementary file 5*), most subjects from the turn-taking dyads (17/20, 85%) realized the coordination benefit, as compared to only half of non-turn-takers (9/18, 50%; Fisher's exact test p=0.035). In addition, 14 of 20 turn-takers (70%) understood the payoff structure, compared to only 1 of 18 non-turn-takers (6%; Fisher's exact test p=0.00005).

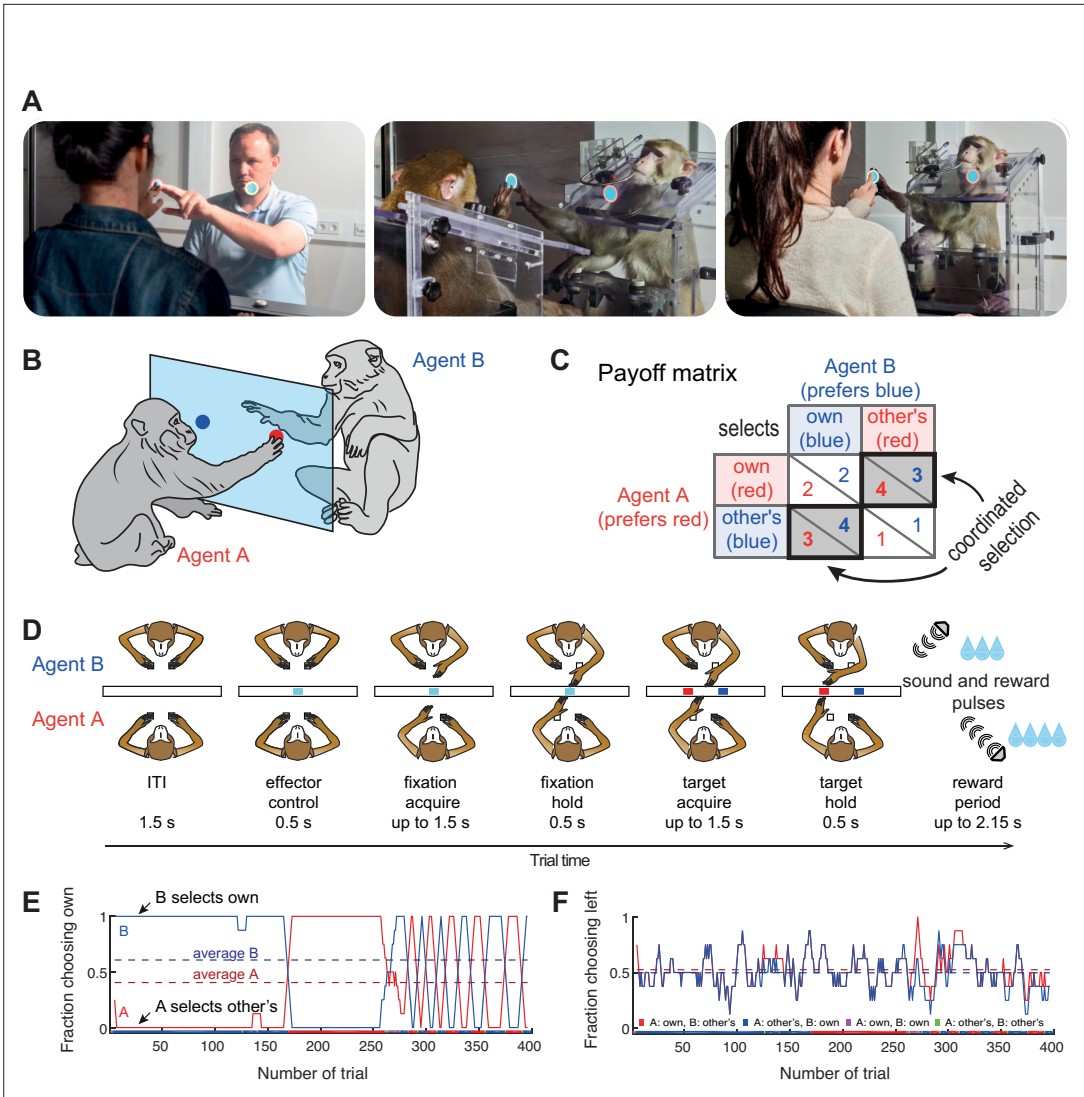

**Figure 1.** Experimental setup. (**A**) Photos of the dyadic interaction platform showing humans and macaques reaching to targets in a shared transparent workspace. (**B**) A sketch and (**C**) the payoff matrix describing reward outcomes for each player for four possible combinations of individual choices (numbers show the monetary/liquid reward units for agent A in red and agent B in blue, 1.5 euro cents per 'unit' for humans and ~0.14 ml of liquid for monkeys). The coordinated selection of the same target (black bold diagonal) resulted in a 'bonus' of two reward units. (**D**) Schematic top-view of two macaque agents in different stages of the task, and the timing of one trial. (**E,F**) Session timecourse of an example human pair (pair 2). (**E**) The fraction of choosing own (FCO) for both agents (A: red; B: blue) as running average over 8 trials. In coordinated trials, one agent always selects the own target (FCO = 1), while the other selects the non-preferred target (FCO = 0). The dashed lines show the session average FCO for each agent. The color bar below shows for each trial the selected combination (red: A and B selected A's preferred target; blue: A and B selected B's preferred target; magenta: both agents selected their individually preferred target; green: both agents selected their non-preferred target). Here both human agents coordinated by selecting the same target in blocks. (**F**) The fraction of choosing left (FCL, from A's viewpoint) as running average over 8 trials. The dashed lines show the session average FCL for each agent. Here, side selection fluctuated around a random level of 0.5 for both players.

All but three pairs achieved an average joint reward above the chance level (2.5 units), and about half the pairs were close to the optimum for the dyad (3.5 units) (*Figure 2C*). Note that turn-taking is not maximizing average joint reward over coordination on a fixed color (in fact, out of the three pairs reaching the maximum average joint reward of 3.5, only one did so by perfect turn-taking, while the other two used static coordination); it does, however, equalize the rewards within a pair. Over all

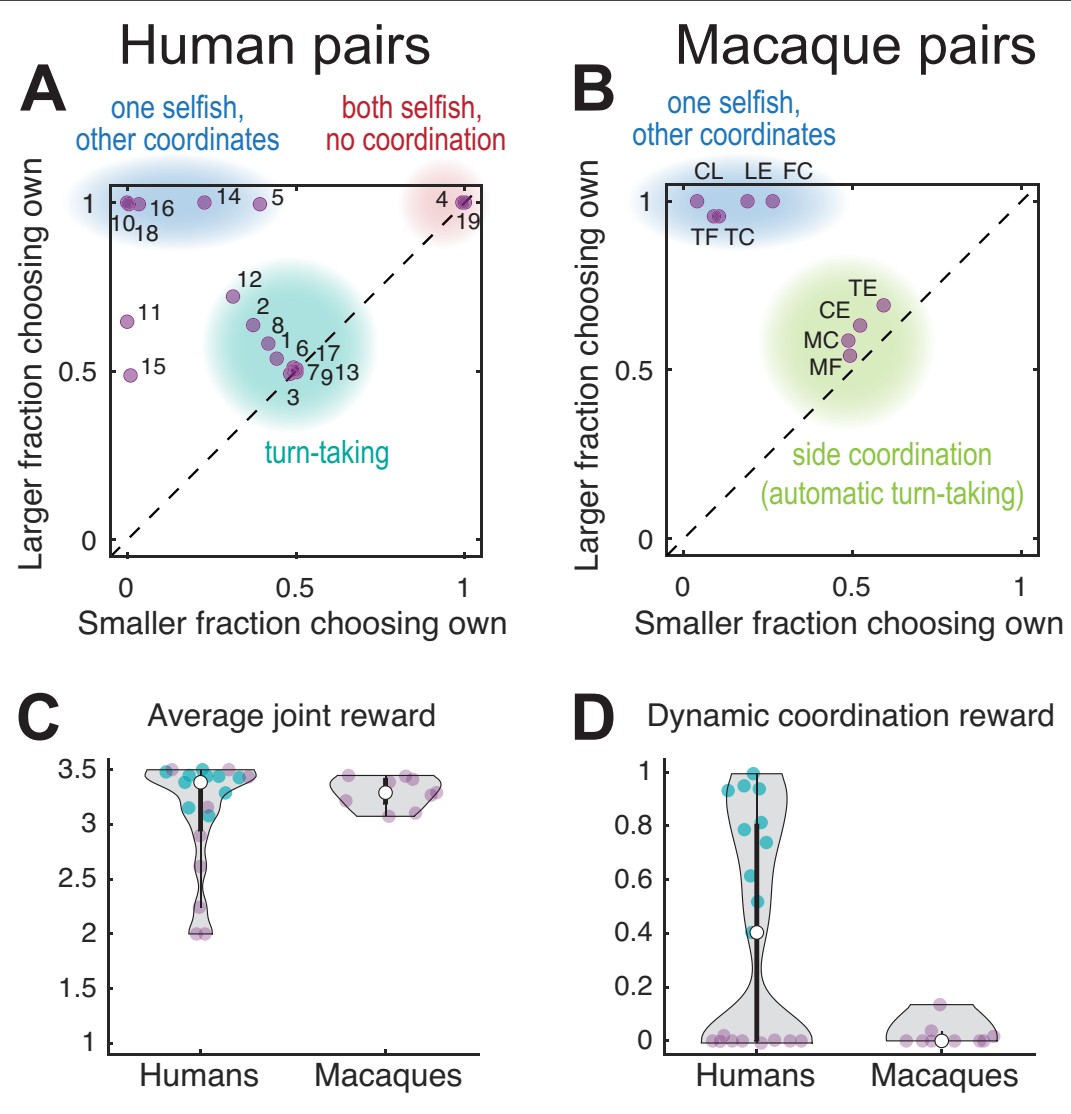

**Figure 2.** Performance of human and macaque pairs. (**A**) Human pairs. The larger of the two average fraction of choosing own (FCO) values in each pair versus the smaller of the two. The diagonal line indicates equal FCO for both agents. Humans converged mainly on three different strategies: balanced turn-taking (teal underlay), static selection of the same color (blue underlay), or static selection of different colors (red underlay); two out of 19 pairs did not converge on any of these strategies. (**B**) Macaque pairs: same as (**A**) for a late session for each pair. Macaque pairs converged on jointly selecting either one of the two color targets (blue underlay) or the same side (green underlay). Note that the clusters within the teal underlay in (**A**) and (**B**) have different underlying choice behavior: turn-taking for humans and fixed side choices for monkeys which results in an automatic alternation of the color targets; to disambiguate these two strategies see *Figure 3*. (**C**) Violin plots of the average joint reward for each pair (purple and teal dots, the latter show 10 turn-taking pairs corresponding to the teal underlay in A); the white dot shows the median of each group. A reward of two indicates uncoordinated choices of the individually preferred targets, a reward of 3.5 indicate perfect coordination, and a reward of 2.5 indicates random independent choices. All but five human pairs and all monkey pairs achieved an average joint reward >3 (no significant difference between species, Wilcoxon rank sum test: human dyads median 3.44, N 14; macaque dyads median 3.29, N 9; rank sum 192.5; p=0.297). Note that the average joint reward cannot disambiguate between balanced and unbalanced rewards for each agent, see *Figure 2—figure supplement 2B and G* for individual rewards. (**D**) The dynamic coordination reward (DCR) for human and monkey pairs. Ten turn-taking human pairs showed a significant DCR with amplitude >0.2 (teal dots, corresponding to the teal underlay in A, see also *Figure 2—figure supplement 2C*) while no monkey pair showed high DCR (only one pair showed a small (<0.2) but significant DCR, see also *Figure 2—figure supplement 2H*).

*Figure 2 continued on next page*

*Figure 2 continued*

The online version of this article includes the following figure supplement(s) for figure 2:

**Figure supplement 1.** Performance improvement in human and monkey pairs.

**Figure supplement 2.** Performance of human and macaque pairs.

**Figure supplement 3.** Exemplary development of choice behavior in macaque pair TE over six sessions.

pairs, the average joint reward increased significantly in the course of the session (*Figure 2—figure supplement 1A*).

To characterize the dynamics of coordinated behavior, we derived a measure of the *dynamic coordination reward* (DCR). The DCR describes the average reward a pair earned *above* (or *below*) the reward expected for independent random choices, given the observed color choice frequencies per agent. Zero DCR indicates coordination by chance, values significantly different from zero indicate above chance levels of dynamic coordination. The DCR values (*Figure 2D*) and the fairly balanced average rewards (*Figure 2—figure supplement 2B*) suggest that 10 of 19 human pairs converged on a form of turn-taking, or reciprocal coordination. These are the same 10 pairs that showed roughly balanced FCO values around 0.5 (teal underlay; *Figure 2A*).

## Macaque pairs converge to simpler coordination strategies

Like humans, macaque monkeys were individually trained to internalize the reward values of the two color targets. All six macaques developed a strong preference for the target associated with the large reward (range 78–100%, mean 95% large reward color selection in the last solo session), before being paired with a conspecific with the opposite target color preference. We then collected the data from nine macaque pairs for multiple sessions to assess on which strategy they converged and how their behavior evolved over time. In the late dyadic sessions, all pairs converged on coordination, in one of two distinct ways: either by converging on the same color target (*Figure 2B*, blue underlay, five pairs) or on the same side (green underlay, four pairs). This side coordination can be seen in high FCL values (*Figure 2—figure supplement 1F*, see *Figure 2—figure supplement 3* for example of developing such coordination in one pair). All pairs achieved above chance average joint reward (*Figure 2C*).

Across all pairs, the average joint reward increased from early to late sessions (median early 2.75 versus late 3.29; p<0.004, two-sided Wilcoxon signed rank test), and in 8 of 9 pairs, the proportion of coordinated trials increased significantly (p<0.05 Bonferroni corrected, Fisher's exact test; *Figure 2—figure supplement 1E*). Indeed, in early sessions there was a clear coordination improvement in the course of the session, leading to an increase in average reward; this effect was not significant in the late sessions likely because macaques already learned to coordinate well at the start (*Figure 2—figure supplement 1C and D*).

## Comparison of behavior between species

To better understand what drives high dynamic coordination in humans (*Figure 2D*), we calculated the *mutual information* (MI) between the sequences of target color choice (MI target) and side choice (MI side) of both agents. These measures indicate how well one agent's choice of color or side can be predicted from the other's choice. Comparing the DCR and the MI measures revealed that high DCR values coincided with non-zero values of *both* MI side and MI color, indicating that dynamic coordination corresponds to both correlated side and correlated color choices between the agents (*Figure 2—figure supplement 2D and E*).

In contrast to humans, monkeys showed either high values of MI color or MI side indicating that the resulting coordination was not achieved by dynamic turn-taking (*Figure 2D*). Instead, the four pairs in the middle of *Figure 2B* (green underlay) that had roughly equal FCO and appeared to be similar to the human turn-takers achieved the coordination by consistently selecting the same side regardless of the target color, which was randomized by the task (Materials and methods).

To directly compare the strategies of human and macaque pairs, we plotted the mutual information values for the side and target color choices of each pair (*Figure 3*). The distance of each pair's location to the origin denotes the strength of (anti)coordination. Locations close to the x-axis denote static side strategies; locations close to the y-axis denote static target color strategies. Ten of 19 human pairs were close to the main diagonal, corresponding to similar values of MI side and MI color.

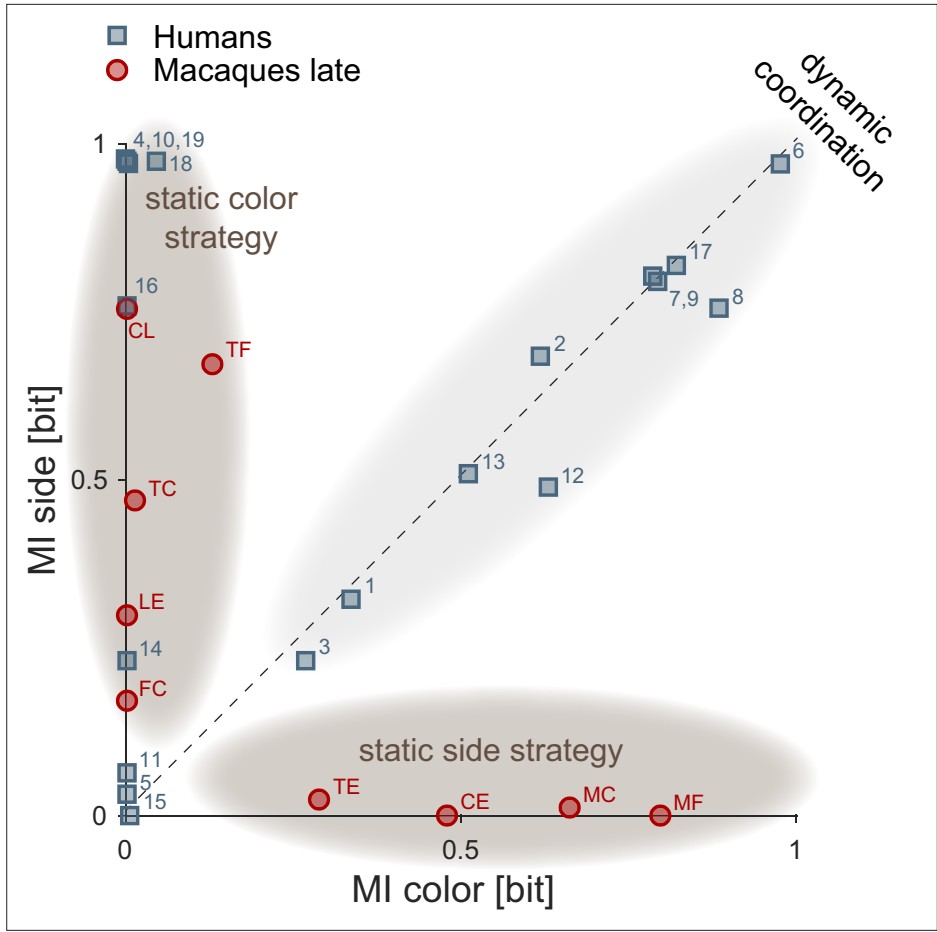

**Figure 3.** Comparison of coordination strategies in humans and macaques. Each symbol shows the mutual information between side choices (MI side) and target color choices (MI color) calculated over the last 200 trials of a session, for human and macaque pairs. Since mutual information for a non-varying sequence of choices is zero, sessions in which a pair statically converged on a fixed color have MI color around zero and a non-zero MI side, and sessions in which a pair statically converged on a fixed side have a non-zero MI color and around zero MI side. Pairs/sessions near the y-axis hence indicate static selection of one of the two target colors (i.e. either coordinating on A's or B's preferred option, or only selecting their own preferred color; dark gray underlays), while pairs/ sessions near the x-axis indicate static coordination on a side. Pairs/sessions with balanced side and color choices fall onto the main diagonal, indicating dynamic coordination or turn-taking (light gray underlay). The distance between each symbol and the (0,0) origin represents the (anti)coordination strength. Ten human pairs showed dynamic coordination, using trial-by-trial or block-wise turn-taking. Eight human pairs converged on color-based strategies: five pairs largely selected one color, i.e., one of the two coordinated options (5, 10, 14, 16, and 18), two pairs did not coordinate (4, 19), and in one pair (11) one agent selected other's color but the other randomly switched between the colors. In the remaining pair (15), both MI side and MI color did not significantly differ from zero (p>0.01, Materials and methods; in this pair one agent mostly selected one side and the other - one color). In late sessions, macaque pairs either converged on selecting the same color target (pairs FC, LE, TC, TF, and CL) or converged on the same side (TE, CE, MC, and MF).

These locations denote trial-by-trial or block-wise turn-taking – a signature of dynamic coordination. In contrast to humans, nine macaque pairs showed simple static side-based (four pairs) or single target color-based (five pairs) coordination, with different levels of coordination strength, but did not employ dynamic coordination.

We next asked whether a difference in action times between the two agents in a pair influenced the coordination by affecting the visibility of each other's actions. Remarkably, all five macaque pairs coordinating on the fixed color converged on the faster monkey's preferred color (*Supplementary file 2*). Likewise, for the static side coordination, all three (out of four) pairs where there was a significant difference in action times between the two agents converged on the faster monkey's less effortful

side closer to the acting arm (*Supplementary file 2*). In humans, such static Leader-Follower relationship also transpired in non-turn-taking dyads coordinating on the fixed color (*Supplementary file 1*), where the faster agent led to its own color and the slower agent followed (e.g., pairs 10 and 14), or the faster agent led to the partner's color and the partner followed (e.g. pair 18). This implies that the static coordination in macaques and some human dyads was shaped by the faster agent's preferences.

## Macaques paired with a human confederate learn to follow

The absence of dynamic coordination in macaque pairs does not preclude the possibility that a slower agent could have monitored the faster partner trial-by-trial in order to coordinate. Alternatively, the faster agent might have reduced the possible outcomes for the slower agent to two options by consistently selecting a fixed side or color, and the slower agent then found the best of the remaining options without resorting to the trial-by-trial following. To test if macaques can actively monitor and adjust to dynamic choice behavior, we trained two monkeys in the dyadic condition with a human confederate who followed a strict pattern of alternating between the two colors in blocks of approximately 20 trials ('confederate training'). If the rhesus monkeys were insensitive to partner's diverse choices, the pattern of the confederate's choices should not affect the monkey's behavior.

Before confederate training, both monkeys adopted either a fixed side or fixed color strategy with the macaque partners (*Supplementary file 2*). When paired with the confederate, both monkeys started with a strong bias for selecting their own preferred color but over time changed to reliable coordination (*Figure 4A, B*, and *Figure 4—figure supplement 1A, B* for another monkey). Their action times (*Figure 4* and *Figure 4—figure supplement 1*, right columns) were initially faster than those of the confederate, and did not vary strongly depending on the confederate's actions. In later sessions, the animals reacted considerably slower in those blocks in which the confederate was selecting the animals' non-preferred color, as if waiting for the confederate to commit to an action. We therefore labeled this as 'following' behavior.

The aggregate measures of choice behavior and coordination development over multiple confederate-paired sessions showed similarities in monkeys' learning (*Figure 4—figure supplement 2*). Both animals reached close to 0.8 DCR units, similar to the values reached by humans employing turn-taking (*Figure 2C*). These results indicate that macaques can take information from their partner into consideration when making decisions in the BoS paradigm. Action time dynamics suggested that the macaques started to actively monitor the confederate's actions and only selected their non-preferred color if the partner started to reach there first.

Due to the repeating block strategy employed by the human confederate, it is not immediately clear whether the monkeys made use of the action visibility and based their decision on the action observed within a trial, or rather on the reward history. We therefore performed a control experiment in which we placed an opaque barrier on the confederate's side of the display for roughly the middle third of a session. The barrier blocked the monkey's view of the confederate hand, while keeping the face visible. In both monkeys, the 'following' behavior ceased and they reverted to selecting their own preferred colors irrespective of the confederate's choices, who – by design – continued the block strategy (*Figure 4C*, *Figure 4—figure supplement 1C*). In line with this observation, there was a significant difference in the monkey's action times when selecting the monkey's preferred (blue) and the non-preferred (red) color in the transparent condition, but no difference in the opaque condition. In other words, monkeys failed to coordinate when they could not see the confederate's actions. Note that reverting to mostly own choices in combination with a block-switching confederate resulted in a reward alternating between two and four units, clearly different from the constant reward of two units in solo trials. Hence, the experienced reward schedule and the visible presence of a confederate render the opaque condition distinct from the solo trials. These results imply that the immediate visual information about partner's actions and not merely monitoring reward history drove and maintained the macaques' 'following' behavior in transparent settings.

## Confederate-trained macaque pair shows dynamic coordination

After two macaques had learned to monitor and follow a partner's actions, we tested how these animals would perform when paired with each other again. At the end of the initial naïve sessions of this pair, and before the training with a human confederate, monkey F (agent A) mostly selected his own preferred color while monkey C (agent B) either selected monkey F's preferred color or the left

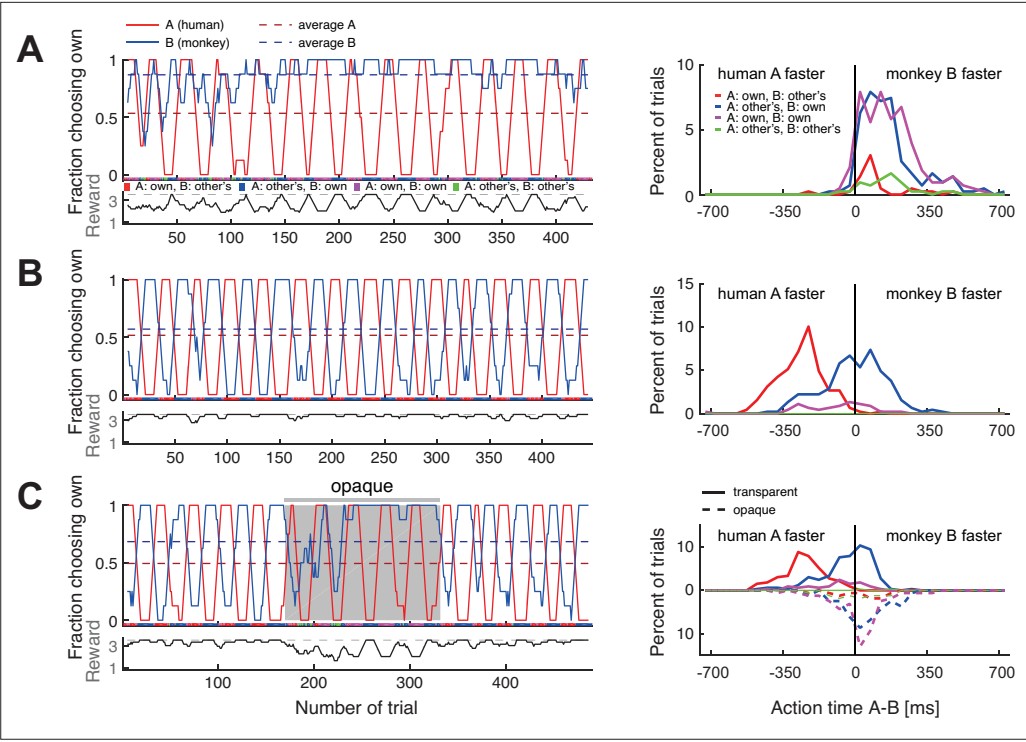

**Figure 4.** Macaques can be trained to take turns when paired with a human confederate. A human partner alternated between the two colors in pre-defined blocks. (**A**) In an early confederate training session, the monkey F (agent B) mostly selected his own target, independent of the human's choices. The lower sub-panel shows the running average (eight trials) of the average joint reward; the cyclic modulation shows that the macaque did not coordinate with the confederate. The action time (AT) difference (AT A – AT B) histograms on the right (bin size 50 ms) show that in all four choice combinations, the macaque acted faster than the confederate. (**B**) A late confederate training session. After several sessions (*Figure 4—figure supplement 2A–F*) the macaque had learned to coordinate (note anti-correlation of the two fraction of choosing own curves, and the average joint reward approaching the maximum 3.5 value). The action time (AT) difference histograms show that in blocks when the pair selected the macaque's non-preferred red target (red curve), the monkey acted slower than the confederate, waiting to see the human's action before committing to his own choice (t-test on AT differences: coordination on red (mean –259 ms, SD 113, N 193) versus blue (mean –7 ms, SD 153, N 217), t(395.5) –19.1, p<0.001). (**C**) Macaque turn-taking with human confederate depended on visibility of the confederate's actions. The gray background denotes the period in which an opaque barrier was placed over the region of the transparent display encompassing all touch targets; hence, the agents could not see each other's reach movements but could still see the face. The macaque switched from turn-taking to mainly selecting his own preferred target during the opaque trials. Upward histograms show the transparent condition (solid curves) and downward histograms show the opaque condition (dashed curves). In the transparent condition, the behavior was similar to (**B**): the monkey waited for the human to act first in blocks of the monkey's non-preferred target (t-test: coordination on red (mean –248 ms, SD 101, N 123) versus blue (mean –39 ms, SD 124, N 166), t(284.5) –15.9, p<0.001). In the opaque condition, the behavior was similar to (**A**): the monkey mostly selected his preferred target and there was no AT difference dependence on human's choices (note the overlap between dashed blue and magenta curves). See *Figure 4—figure supplement 1* and *Figure 4—figure supplement 2G–L* for monkey C.

The online version of this article includes the following figure supplement(s) for figure 4:

**Figure supplement 1.** Macaques can be trained to take turns when paired with a human confederate.

**Figure supplement 2.** Monkeys gradually learn to follow a human confederate's turn-taking behavior.

---

side. At the end of the confederate training, both F and C followed the confederate. The question was: would they now express turn-taking behavior or would they fall back to their initial static color or side coordination behavior?

The pair's coordination behavior changed noticeably after training with a confederate. In the first session, after some initial back and forth, the pair converged on mostly selecting B's color, with B acting faster (*Figure 5A*). The most interesting phase was during the second session, in which the

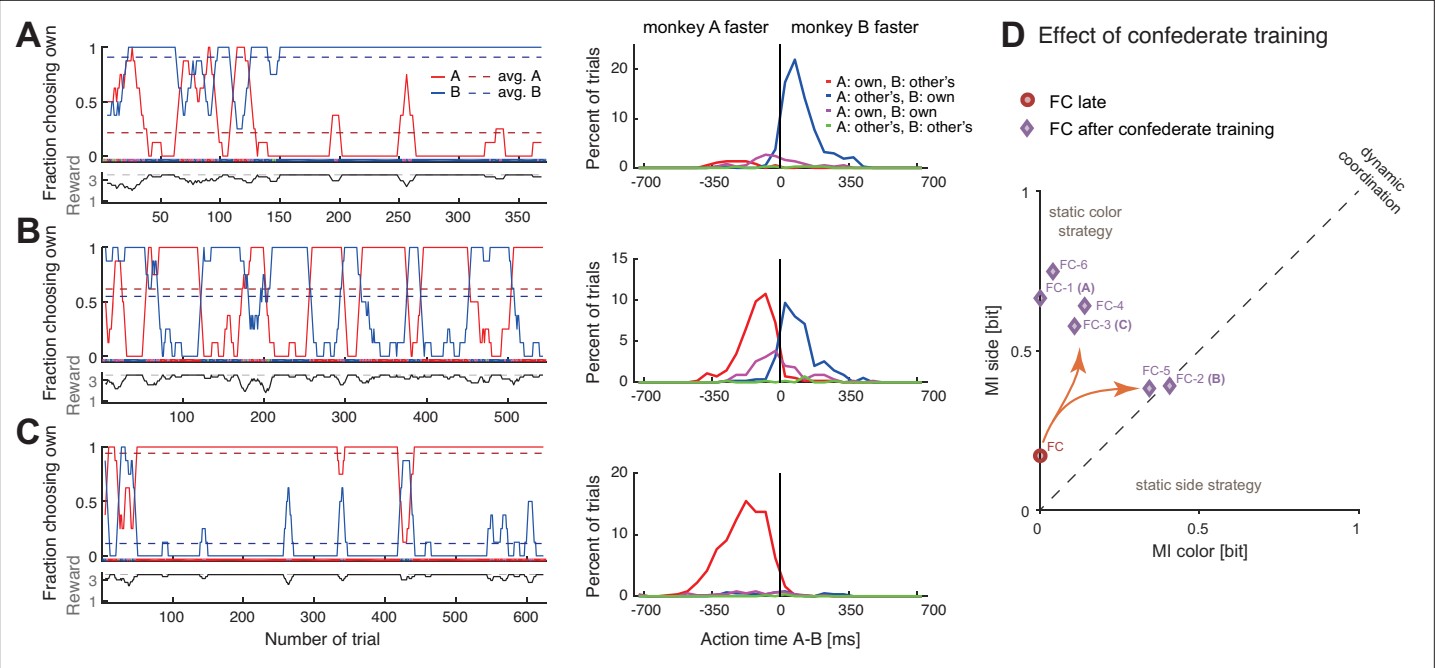

**Figure 5.** Confederate-trained macaques show competitive turn-taking behavior. (**A**) In the first paired session after confederate training (FC-1) indications of turn-taking followed by a convergence on agent B's target can be seen in the fraction of choosing own curves in the top subpanel; note how agent A occasionally tried selecting his preferred target. The action time (AT) difference histograms show that agent B acted faster when both agents selected his preferred target (t-test on AT differences: coordination on red (mean –211 ms, SD 112, N 27) versus blue (mean 105 ms, SD 98, N 286), t(29.9) –14.1, p<0.001). (**B**) In the next session (FC-2) the pair developed turn-taking; the action time (AT) difference histograms show that the pair converged on the faster monkey's target (t-test: coordination on red (mean –117 ms, SD 96, N 238) versus blue (mean 90 ms, SD 104, N 201), t(411.7) –21.4, p<0.001). (**C**) In the following session (FC-3), the pair mostly converged on A's preferred target, with A now acting faster (t-test: coordination on red (mean –188 ms, SD 115, N 553) versus blue (mean –31 ms, SD 192, N 33), t(33.4) –4.6, p<0.001); note that as in the first session the agent B occasionally tried his own preferred target, and two of these attempts were reciprocated by A. (**D**) Change in coordination in pair FC from a late session (red circle), to six sessions after confederate training. The 'coordination strength' (distance from the origin) increased, and in two sessions there was a significant dynamic coordination (FC-2, FC-5).

The online version of this article includes the following figure supplement(s) for figure 5:

**Figure supplement 1.** Confederate-trained macaques.

animals repeatedly alternated between the two colors in long blocks (*Figure 5B*). In the third session, the pattern was opposite to the first session, with the pair converging mostly on A's color who acted faster (*Figure 5C*). This time, B tried occasionally to switch to his preferred color, and A followed for three short periods. The mutual information measures as well as the DCR values indicate that after confederate training, the coordination strength increased and dynamic coordination resulted in turn-taking behavior in two sessions (*Figure 5D*, *Figure 5—figure supplement 1*).

If action visibility is driving this form of dynamic coordination, then the question of who successfully 'insists' on own preference and who 'accommodates' the other's preference should depend on the relative action times of the two agents. Indeed, in the session in which the animals exhibited apparent turn-taking behavior (*Figure 5B*), the faster animal (Leader) selected the preferred color, while the slower animal (Follower) had to accommodate. Thus, the dynamic coordination arose within a competitive interaction.

## Comparison of dynamic coordination between species

At first glance, the turn-taking in the confederate-trained macaques resembled the long turn-taking observed in some human pairs (*Figure 5D* with *Figure 3*). Yet, the analysis of action time distributions suggests that macaques employed a competitive version of turn-taking in which the faster agent selects its own color (insists) while the slower agent follows (accommodates). Such behavior would be in line with theoretical predictions derived from our evolutionary simulation (*Unakafov et al., 2020*), showing that competitive turn-taking provides the most effective strategy for a BoS type game when

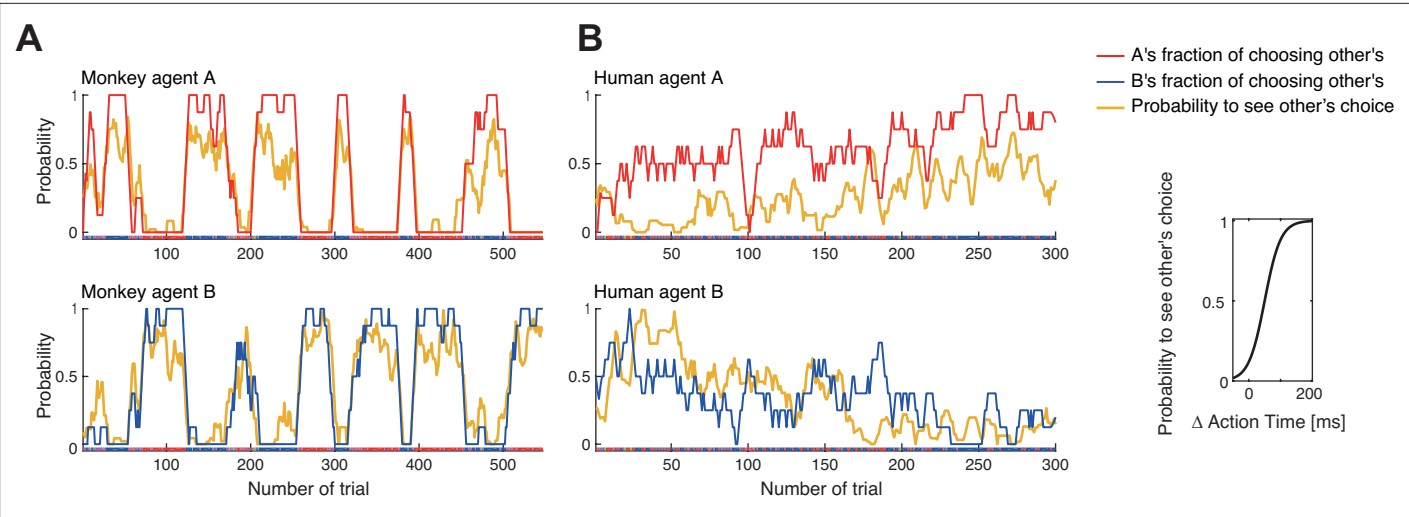

**Figure 6.** Confederate-trained macaque turn-taking is driven by action visibility. The inset on the right shows the logistic mapping function from the action time difference to the probability to see the partner's choice before acting. (**A**) The 2nd session of confederate-trained macaque pair FC (also shown in *Figure 5B*). This panel shows the correlation between the probability of seeing the other move first (the probability to see other's choice) and the probability of following to the other's preferred target (the fraction of choosing other's). The fraction of choosing other's is the inverse of the fraction of choosing own. All curves show running averages of the respective measure over eight trials. For agent A the correlation coefficient r(545) was 0.92 and for agent B 0.91 for smoothed data, and 0.64 and 0.65 for non-smoothed (trial-by-trial) data (all p<0.001). (**B**) The human pair 12 with the highest positive correlation showed a different pattern: a slow gradual shift from B following A to B leading A over the course of the session. For A, the correlation coefficient r(298) was 0.52 and for B 0.46 for smoothed data, and 0.27 and 0.33 for non-smoothed data (all p<0.001). The same human pair (only one out of ten pairs that exhibited dynamic coordination) showed a significant difference between action time (AT) differences in coordinated choices favoring agent A versus agent B, with each agent being faster when selecting the own preferred color (t-test on AT differences: coordination on A's red color (mean –88 ms, SD 129, N 96), coordination on B's blue color (mean 12 ms, SD 109, N 180), t(168.8) –6.5, p<0.001). See *Figure 6—figure supplement 1* for data on other human pairs and on other sessions of macaque pair FC.

The online version of this article includes the following figure supplement(s) for figure 6:

**Figure supplement 1.** Correlations between the probability to see the partner's choice and the likelihood to select the partner's target.

players have a high probability to observe partner's choices. To further test the hypothesis that this macaque pair developed a competitive Leader-Follower strategy, we performed a correlation analysis between the modeled visibility of the faster agent's action by the slower agent and the observed likelihood of following to the faster agent's color (the faster animal nearly always selected his own color). *Figure 6A* shows macaque agent A's probability to select his non-preferred color, the 'fraction of choosing other's' (the inverted fraction of choosing own) and the modeled probability for A to see B's choice before A made own choice, and the same for macaque agent B (see Materials and methods for details on modeling). In both monkeys, there was a strong correlation between the model and the observation, ranging from 0.92 for smoothed data to 0.64 for unsmoothed data, all significant at p<0.001. This analysis further supports the notion that the confederate-trained macaque pair had developed a competitive turn-taking (alternating Leader-Follower) strategy. Strong positive correlations between action visibility and 'following' behavior were present in all six post-confederate-trained sessions, indicating bouts of dynamic coordination (*Figure 6—figure supplement 1*; *Supplementary file 3*).

In contrast, human participants did not show a strong trial-by-trial relationship between relative action time differences and choices (*Supplementary file 4*). Even in the pair with the strongest positive correlations (pair 12, *Figure 6B*), the 'following' behavior was weak and resulted from a gradual transition from B initially following A (to both colors) to A following B (predominately to B's color) later in the session, rather than from alternating competitive turn-taking. The lack of such a correlation does not mean that action times and action visibility played no role in human choices. Indeed, in 47% (9 of 19) of human pairs, the absolute difference in action times between the agents significantly differed in coordinated trials compared to uncoordinated trials (*Supplementary file 1*). Furthermore, turn-taking dyads had a larger absolute action time difference (102.7 ± 38.7 ms, N 10) than non-turn-taking dyads (61.3 ± 21.6 ms, N 9; t-test correcting for unequal variance t(14.4) 2.918; p=0.011); this difference was

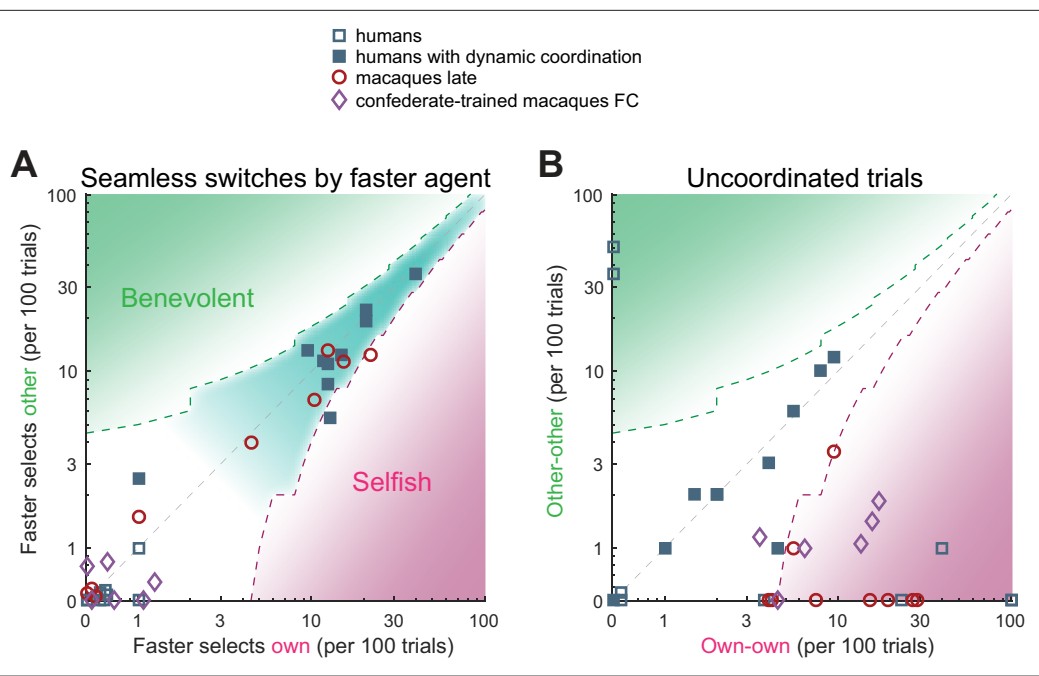

**Figure 7.** Humans and macaques show different switching behavior. A quantification of different transition aspects as scatter plots of benevolent choices on the y-axis and selfish choices on the x-axis, in the two species. (**A**) The behavior of a faster agent for all seamless switches between the two coordination modes (red-to-blue and blue-to-red). Here and in B, the x-axis shows the number of own target selections (selfish), the y-axis the number of non-preferred target selections (benevolent). Turn-taking humans as well as monkey pairs that coordinated on one side showed high and balanced numbers of seamless switches, while the confederate-trained macaque pair and the non turn-taking humans showed only a few seamless switches. The dashed contours show the 95% CI around the diagonal, the teal underlay shows the area with balanced selfish and benevolent choices, the green and the magenta underlays show the area with significantly more benevolent and selfish choices, respectively. (**B**) The number of uncoordinated trials in which both subjects selected their respective preferred color (own-own) on the x-axis, and the number of trials in which both subjects selected their non-preferred color (other-other) on the y-axis. Turn-taking human pairs mainly lay along the diagonal while macaque pairs cluster in the lower right region (mostly significantly selfish).

The online version of this article includes the following figure supplement(s) for figure 7:

**Figure supplement 1.** Humans and macaques show different switching behavior.

**Figure supplement 2.** Action time changes around choice combination switches.

mostly driven by the coordinated trials (105.0 ± 41.5 ms, N 10 versus 64.7±24.1 ms, N 7; t(14.7) 2.523; p=0.024), while a similar trend in trials without coordination did not reach significance (93.8 ± 29.2 ms, N 10 versus 69.5 ± 39.7 ms, N 9; t(14.6) 1.512; p=0.152). This implies that turn-takers tended to pay more attention to partner's actions and wait more for each other during the coordination. Hence, similarly to macaques, seeing the faster agent's choice can help the slower agent to coordinate. But unlike the confederate-trained macaque pair, dynamic human coordination was not driven by the faster agents striving to select only the own preferred color (except human pair 12 shown in *Figure 6B*): for instance, in the turn-taking pairs 3 and 17, the faster agent B led both to own and A's preferred colors, while A followed (*Supplementary file 1*).

The action time analyses show that for the confederate-trained macaques, the visibility of the other's action before making one's own decision correlated with the probability to follow the choice of the other, with the faster monkey 'selfishly' selecting its own preference. Turn-taking in humans, however, did not rely on temporal competition. To highlight this difference further, we asked how the two species maintained the coordination. Macaques transitioned between periods of coordination using short periods of non-coordination, while turn-taking humans tended to switch seamlessly between the two coordination options without verbal communication (*Figure 7—figure supplement 1A and B*). Turn-taking humans showed a high and roughly equal amount of benevolent and selfish

seamless switches by a faster player (*Figure 7A*, teal underlay). The four macaque pairs showing the highest number of seamless switches used a side-based strategy, which due to the color-to-side randomization trivially generated seamless color switches. Confederate-trained macaques and non-turn-taking humans showed only very few seamless switches.

Alternative to seamless switches, agents can end the current coordination block by switching to non-coordination. These trials can be used to initiate or 'signal' a switch of coordination. When turn-taking humans showed uncoordinated trials they lay along the unity diagonal, indicating a benevolent intention to initiate the switch to the other's preferred color in half of these trials (*Figure 7B*). All macaque pairs were below the unity diagonal, indicating a selfish preference for own color. In the confederate-trained macaques, the coordination epochs were separated by epochs of non-coordination that looked like each agent 'challenged' the other to accommodate. Indeed, during challenge initiations and resolutions, the faster macaques tended to act selfishly, while turn-taking humans showed balanced switching behavior (*Figure 7—figure supplement 1C and D*). The analysis of action times in confederate-trained macaques showed that the faster agent slowed down and the slower agent sped up when transiting to non-coordination, and conversely, the insisting agent sped up to initiate the coordination on the own color and the accommodating agent slowed down (*Figure 7—figure supplement 2*).

In sum, there were no indications of competitive turn-taking in humans, as the faster agents showed a balanced selection of both coordination options. Furthermore, in four out of the ten turn-taking pairs, one agent was significantly faster than the other in coordinated trials as compared to uncoordinated trials, regardless if it was coordination on its own or the other's color (*Supplementary file 1*). This means that such an agent initiated switches to and from the own color; while being able to see the faster agent's action likely helped the slower agent to maintain coordination and to accomplish seamless switches. In contrast, the confederate-trained macaques showed no seamless switches, but displayed temporal competition in which the faster agent led to his own color.

## Discussion

We studied macaque and human pairs in a coordination game, which offered higher rewards for selecting the same option but entailed an inherent conflict about which of the two coordinated options to select. Both species largely converged on coordinated behavior but in a markedly different fashion. Half of the human pairs achieved nearly optimal coordination and fair (cooperative) turn-taking that equalized the rewards of the two partners, without explicit communication. Macaques, instead, exploited simpler strategies that maximized reward without the need to track the actions of the partner in real-time. That is, they used static instead of dynamic coordination. After two macaques were trained to observe and attend to the partner's choice with a human confederate, their behavior shifted to competitive turn-taking. In post-confederate-training sessions, the choice behavior was strongly correlated with trial-by-trial differences in the reach action times: the faster monkey chose his preferred option and the slower one followed. Our results show that both humans and macaques make use of action visibility by taking information about the other's action into account before making their own decisions. There was, however, a fundamental difference: when coordinating dynamically, monkeys showed competitive turn-taking while in humans the faster agent often 'offered' switching to the partner's preferred option, exhibiting a form of benevolent and fair turn-taking.

### Coordination in humans and macaques

Ten of the 19 human pairs converged on dynamic coordination that balanced the reward, developing fair, cooperative turn-taking, while another five pairs coordinated on the same fixed color throughout the session. Similar to the approach of Brosnan and colleagues (*Brosnan et al., 2017*; *Brosnan et al., 2011*), the human subjects in the present study had to infer the underlying payoff rules while playing, closely matching the procedure in macaques. Even under such conditions, the majority of humans arrived at near-optimal behavior within a single session, in our study as well as in other coordination and anti-coordination games (*Brosnan et al., 2017*; *Brosnan et al., 2012*; *Brosnan et al., 2011*). The observed turn-taking frequency was close to other variants of BoS games with explicit instructions of the payoff matrix, where approximately 50–60% of human subjects developed turn-taking (*Arifovic and Ledyard, 2018*; *Ioannou and Romero, 2014*; *Melis et al., 2016*; *Sonsino and Sirota, 2003*).

What sets the turn-takers apart from the dyads that coordinated on a fixed color, or did not coordinate well? Based on their debriefing responses, turn-takers displayed a much better understanding of the implicit reward rules such as the value of coordination and unfairness. The non-turn-taker group was more diverse: some subjects did not understand the game or only understood some aspects; other subjects had a better understanding but were selecting selfishly or could only follow the partner. Furthermore, the turn-taking dyads utilized action visibility driven by action time differences more extensively, although some non-turn-taking dyads overall also exploited action visibility to coordinate on a fixed color.

In contrast to humans, macaque pairs converged on simpler strategies, either selecting the same color (56%) or selecting the same side (44%). These two strategies require less cognitive resources and trial-by-trial coordination effort than turn-taking, because they allow each agent to make decisions without paying attention to the trial-specific actions of the partner. Although we found that all macaque pairs that exhibited a significant difference in action times between the agents (8 of 9 pairs) converged on the faster agent's preferred choice, it seems that initially, macaques did not rely on trial-by-trial action visibility. This is suggested by the fact that it required relatively long training before macaques started to follow a human confederate dynamically. Instead, we speculate that slower macaques adapted to their partner's static choice by maximizing their reward over the remaining options.

However, all but one animal coordinated both via fixed color or fixed side selection depending on the specific partner, and no animal persisted on the own color with all tested partners. For instance, monkey F insisted on his preferred color when playing with monkey C, accommodated to the other's color when playing with monkey T, and converged on his right side when playing with monkey M (*Supplementary file 2*). This result indicates that macaques took the partner's actions and reward outcomes into account, albeit not necessarily in a strict trial-by-trial or within-trial fashion, resembling the formation of stable color selection conventions in Guinea baboons in a simple sequential coordination task (*Formaux et al., 2022*). For instance, after a few sessions of selecting mainly own color, one agent changed either to selecting their non-preferred color (increasing his reward to 3 and the other's to 4 drops) or to selecting one side (increasing his reward to an average of 2.5 and the other's to 3 drops). If the other monkey also started favoring the same side, this increased both agents' reward to 3.5. This sequence of steps illustrates how macaque pairs can develop their strategy over the course of several sessions.

While our study is, to the best of our knowledge, the first implementation of a BoS paradigm in macaques, other 2×2 games have been investigated in several primate species (humans, chimpanzees, macaques, and capuchins). So far, the data suggest that monkeys, unlike humans, do not spontaneously adopt turn-taking, despite coordinated reciprocity being a successful evolutionary strategy in simulated non-communicating agents (*Colman and Browning, 2009*). For instance, in the anti-coordination Conflict game (*Brosnan et al., 2017*), capuchins and macaques converged on only one of the two possible asymmetric equilibria, much like 4 of our 9 macaque pairs (while many human pairs balanced payoffs by alternating between the two equilibria). One can argue that turn-taking is not expected in macaques, since it has been suggested that even apes don't exhibit it. In an exceptionally stringent variant of a BoS-type game, in which coordinated rope pulls were required to deliver a reward only to one of the two agents (payoff matrix zero except 0/1 or 1/0 at the coordinated diagonal), chimpanzees, unlike human children aged 5, did not converge on trial-by-trial alternation/turn-taking (*Melis et al., 2016*). This difference was interpreted as a species difference in the strategic capability to maintain longer-term cooperation (e.g. by motivating the other to reciprocate). Nevertheless, apes cooperated in 64% of trials, and in the examples of ape dyad data, one can see some switching between the agents' choices, albeit not trial-by-trial but in longer blocks. Given that even humans often balance the fairness over longer time frames, as we observed in several human dyads (e.g. *Figure 1E*), it was important to test if also macaques might exhibit a similar block-wise turn-taking under conditions of more trials than ape experiments and less conflict (i.e. when both agents get increased reward for coordination).

## Action visibility and dynamic coordination

The lack of spontaneous dynamic coordination such as turn-taking in naïve macaques in our experiment cannot be attributed to their inability to infer the upcoming choices of the partner, because they

could observe the partner's actions through the transparent display. Especially for nonhuman species, coordinating based on mutual choice history might be more demanding than coordinating using the immediately observable behavior of others. Coordination (or anti-coordination) typically is improved with increasing information about the other's choice by shifting from the strictly simultaneous to the sequential mode (*Brocas et al., 2018*; *Rapoport, 1997*). The transparent game approach we adopted here differs from those two classical modes in that each subject can decide independently when to act (within a certain time window). Under such conditions, the coordination in chimpanzees is facilitated if one of the agents consistently acts faster than the partner (*Bullinger et al., 2011*). Compared to an opaque simultaneous setting, visual feedback about the partner's choices improved macaques' and capuchins' coordination in a transparent (called asynchronous) versions of a Stag Hunt game (*Brosnan et al., 2012*). Similarly, there were substantial differences in capuchins' and rhesus' behavior in the Conflict/Chicken game when they had access to the current choice of a partner (*Brosnan et al., 2017*; *Ong et al., 2021*).

In the studies of Brosnan and colleagues, subjects looked at the same split monitor while sitting next to each other, and could observe each other's decisions as cursor movements. In our setup, agents sat opposite to each other and saw actual eye, head and hand movements, combining face-to-face and action visibility. In humans, face-to-face visibility in a simultaneous iterated Prisoner's Dilemma (iPD) and in the Ultimatum game significantly improves mutual cooperation, even without action visibility (*Jahng et al., 2017*; *Tang et al., 2016*). These findings support the idea that non-verbal social signals and, more generally, the observable presence of others influence decision-making. The ongoing action visibility per se also had a profound effect on mutual cooperation in iPD (*Friedman and Oprea, 2012*) and in a web interface-based anti-coordination game conceptually similar to BoS (*Hawkins and Goldstone, 2016*). In a competitive reaching task, a face-to-face transparency allowed human subjects to glean useful information from observing the relevant hand effector and from seeing the face and the full body of an opponent (*Vaziri-Pashkam et al., 2017*). In our study, action visibility facilitated sensorimotor coordination, as demonstrated by action time differences in transparent vs. opaque conditions, convergence to a faster agent's preference during static coordination, a tight relationship between action visibility and choice behavior in turn-taking macaques, and inter-agent action time differences in coordinated vs. non-coordinated trials. Seeing actual movements rather than relying on abstract representations of others such as cursor motions might be especially crucial in nonhuman experiments. But beyond immediate sensorimotor aspects, acting face-to-face on the same targets in close proximity likely influenced social cognition on a more general level.

The importance of action visibility is also supported by computational modeling. In the continuous-time cooperation games such as iPD and Stag Hunt where agents could observe and respond to each other's actions in real-time, cooperation by coaction is more easily obtained and stabilized against exploitation than the cooperation that relies on delayed reciprocity (*van Doorn et al., 2014*). For the iPD and BoS transparent games, we showed that different coordination strategies are preferable for different probabilities of seeing partner's choice (*Unakafov et al., 2020*; *Unakafov et al., 2019*). When these probabilities are low (or when agents do not utilize the action visibility), simple strategies like 'win-stay, lose-shift' are most effective. For higher probabilities of seeing partner's choice more complex strategies emerge, such as coordinated turn-taking and temporal Leader-Follower, where the faster agent determines the choice of the slower agent. For the selfish agents, this means that the faster one insists on its individually-preferred own target and the slower accommodates (*Unakafov et al., 2020*).

Given that many human pairs exhibited spontaneous dynamic coordination, while naïve macaques did not, we can ask if and how these two species utilized the action visibility. Both species exploited action visibility to achieve and/or maintain the turn-taking, emphasizing the importance of the 'transparency' of interactive behavior as an important determinant of emerging strategies. Humans relied on action visibility for seamless switches from one color to another, and for maintaining coordination within a block of trials. But besides one pair, there was no indication that humans employed the competitive variant of a Leader-Follower strategy. Instead, faster agents were as likely to switch from their own to the other's preferred color, indicating benevolent turn-taking. These results add to the body of literature indicating that humans in a social setting might base their decisions not only on pure reward maximization (*Clavien and Chapuisat, 2016*). Such normative behavior can also

reflect prospective planning to avoid conflict and maintain stable individually-beneficial cooperation in subsequent trials.

Very different dynamics transpired in additional experiments with macaques. Pairing them with a human confederate who consistently alternated between colors in short blocks but did not intentionally provide any additional social cues demonstrated that macaques are capable of closely follow partner's actions (*Nougaret et al., 2019*), and adopt the imposed 'turn-taking' to maximize their reward. Moreover, temporarily blocking the view of the confederate's hand abruptly changed macaque behavior, disrupting the coordination. The macaques' coordination was thus driven by action visibility and not by strategies like 'win-stay, lose-shift' or trial counting. In line with the Leader-Follower strategy, macaques were faster than the confederate when selecting own color, but waited for the confederate when selecting the confederate's color. Intriguingly, pairing the two macaques that completed the training with a human confederate resulted in behavior different from both naïve macaque behavior and from the 'following behavior' with the confederate. These macaques established competitive dynamics, with the faster agent selecting his preferred color and the slower agent following, as predicted by our evolutionary simulations (*Unakafov et al., 2020*). Such competition resulted in either sustained coordination on one fixed color or in turn-taking. Comparing the temporal signatures of the action time differences in trials around a switch to or from coordinating on a specific color further indicated that macaque turn-taking was competitive and dynamic in nature. The break of coordination was triggered by the faster agent slowing down and the slower one speeding up and selecting his preferred color; conversely, the transition to coordination was associated with the speeding up agent selecting his preferred color and the slowing down agent accommodating.

Notably, we considered the interactions in the pairs where one monkey underwent the confederate training, and other did not (*Supplementary file 2*). In all four instances, static strategies prevailed, suggesting that turn-taking as a joint strategy behavior requires active participation of both agents. We note, however, that in all three cases of fixed color strategy, the confederate-trained monkeys accommodated the selfish choice of their partners, perhaps because they had learned to monitor and follow partner's actions.

Species differences in general cooperative behavior may explain both the reluctance of naïve macaques to coordinate dynamically, as well as the competitive nature of dynamic interactions when they occur. The transparent BoS game requires the agents to cooperate in close proximity in the pursuit of an immediate reward, which necessitates some degree of social tolerance by both partners. If we take natural food-sharing behavior as a proxy for social tolerance, it is not equally prevalent among primate species. Humans, apes, some baboon species (*Goffe and Fischer, 2016*), and some New World monkey species including capuchins share food between adults, while rhesus macaques do not even share food with their offspring (*Jaeggi and Gurven, 2013*). In line with these patterns, mutually beneficial alternating task performance and turn-taking have been observed in humans (*Brosnan et al., 2017*; *Grueneisen and Tomasello, 2017*; *Helbing et al., 2005*; *Rapoport et al., 1976*), apes (*de Waal, 1989*; *Martin et al., 2017*), and capuchins (*Parrish et al., 2015*), but not in more despotic, less tolerant rhesus macaques (*Joly et al., 2017*).

Beyond social attitudes, the lack of the cooperative turn-taking in nonhuman primates might reflect cognitive limitations in long-term planning and perspective-taking (*Stevens and Hauser, 2004*). In humans, performance/level of coordination in BoS and other iterative games was shown to depend to a larger degree on intelligence (to form good strategies) than on personality or social predispositions (*Proto et al., 2019*). While the action transparency decreases the reliance of coordination on mental representations, the differences between strategies in macaques compared to humans might still be driven by more limited cognitive capacities.

## Limitations and future directions

There are several limitations in our study that have to be considered and possibly addressed in future experiments. First, we only tested six macaques (and nine pairs), all males, and only 1 pair was housed together (others were from neighboring but separate enclosures). Therefore, we cannot say much about the influence of social rank. Anecdotally, however, it did not seem that the more dominant agent always prevailed. For instance, one animal (monkey T), the smallest and likely to be subordinate to all other partners, successfully 'insisted' on his own color in two out of three cases (*Massen et al., 2010*). Furthermore, we only tested two confederate-trained macaques so far. While the results are

exciting and show that macaques can engage in dynamic competitive turn-taking, future experiments will need to test how generalizable this pattern is.

Second, due to the task target color/side randomization, selecting targets on the same side of the display resulted in low effort, efficient and fair coordination. We cannot say if fairness was a contributing factor in convergence of the four macaque pairs to this strategy, although we deem it highly unlikely. A new experiment where the probability of a specific color to appear on the same side is parametrically modulated is needed to evaluate the conditions under which this strategy emerges (or what level of unfairness is accepted). Furthermore, we prompted the human participants to pay attention to the partner's choices, to facilitate the emergence of coordination within a single session. It would be interesting to test human pairs in several sessions and without any instructions, similar to macaques, to see if humans might also converge on the fixed side strategy as means of fair and seamless coordination. It requires preventing the human participants from any verbal contact after and between the sessions, however.

Third, it would be important to vary the ratio and the range of reward magnitude ('stakes'), as it has been shown that stakes may strongly affect the individual (*Schultz, 2015*) and social (*Hawkins and Goldstone, 2016*) decisions. Inherently, the nature of reward differed between the species: humans received monetary compensation while macaques worked for a liquid reward. Note, however, that in each session macaques were offered an opportunity to work and obtain more liquid in a solo regime after the dyadic experiment (Materials and methods), so the participation in a dyadic interaction was not forced, and the end of the dyadic sessions was not associated with a fully waning motivation. We also did not observe that variations in satiation level in the course of each session played a substantial role in shaping macaque strategies. Nevertheless, it is important to acknowledge differences between primary and secondary rewards, and potential divergence in the underlying motivations and learning, beyond the species differences in socialization and cognitive capacities considered above.

Finally, it would be very interesting to relate diverse in-game behavioral strategies observed in the human pairs to individual differences in cognitive and socio-economic personality traits (*Rollwage et al., 2020*; *Zhao and Smillie, 2015*), and link them to stable behavioral phenotypes (*Peysakhovich et al., 2014*; *Poncela-Casasnovas et al., 2016*). This, however, will require a much larger sample and personality-oriented study design.

In summary, we show that both, humans and nonhuman primates, dynamically coordinate in real-time transparent interactions, but macaques compete while humans strive for fairness by prosocial turn-taking. Enabling action visibility with our transparent setup provides novel comparative insights into social decision processes as they unfold in real-time during transparent interactions, and offers a new route to behavioral and neural investigations of dynamic decision-making in cooperative and competitive contexts (*Ferrari-Toniolo et al., 2019*; *Isoda et al., 2018*; *Yoo et al., 2021*).

## Materials and methods
### Participants
#### Humans

38 right-handed subjects (23 females, mean age: 26.1 ± 4.1 SD, range 20–41 years) participated in the study as paid volunteers. Subjects were tested as 19 unique pairs, i.e., each participant contributed only once. Instructions given to the subjects before the experiment are provided in Appendix 1 (*Supplementary file 5*). In short, subjects were instructed on how to operate the setup and to interpret the auditory feedback as an indicator of the earned reward. They were not given an explicit description of the task's payoff structure beyond 'Your reward will depend on your own and your partner's choice.' Prior to the experiment, subjects were individually familiarized with the setup and practiced a single-player ('solo') version of the task. All subjects gave written informed consent for participation after the procedures had been explained to them and before taking part in the experiment. Experiments were performed in accordance with institutional guidelines for experiments with humans and adhered to the principles of the Declaration of Helsinki. The experimental protocol was approved by the ethics committee of the Georg-Elias-Mueller-Institute for Psychology, University of Goettingen (GEMI 17-06-06 171). We excluded two additional pairs from the analyses due to experimental differences (one pair did not perform the initial individual training; one pair ended the experiment prematurely).

## Macaques

Research with nonhuman primates represents a small but indispensable component of neuroscience research. The scientists in this study are committed to the responsibility they have in ensuring the best possible science with the least possible harm to the animals (*Roelfsema and Treue, 2014*). The experimental procedures were approved by the responsible regional government office (Niedersaechsisches Landesamt fuer Verbraucherschutz und Lebensmittelsicherheit (LAVES), permits 3392-42502-04-13/1100 and 3319-42502-04-18/2823), and were conducted in accordance with the European Directive 2010/63/EU, the corresponding German law governing animal welfare, and German Primate Center institutional guidelines.

Six adult male rhesus monkeys (designated by initials C, E, F, L, M, and T) participated in the study, yielding nine pairs (each monkey participated in two or more pairs). Animals were extensively trained with positive reinforcement to climb into and stay seated in a primate chair. The animals were pair- or group-housed in facilities of the German Primate Center (DPZ) in accordance with all applicable German and European regulations. The facility provides the animals with an enriched environment, including a multitude of toys and wooden structures (*Berger et al., 2018*; *Calapai et al., 2017*), natural as well as artificial light and access to outdoor space, exceeding the size requirements of European regulations, and a rich diet including primate biscuits, fruit, and vegetables. During the study, the animals had unrestricted access to food and fluid, except on the days where data were collected or the animal was trained on the behavioral paradigm. On these days, the animals were allowed access to fluid through their performance in the behavioral paradigm. The DPZ veterinarians, the animal facility staff, and the lab scientists carefully monitored the welfare of the animals.

## Experimental setup

For maximal comparability of the human and the monkey behavior, we developed a novel dyadic interaction platform in which two human or nonhuman primate subjects co-act in a shared workspace while sitting face-to-face (*Figure 1A*). Joint dyadic tasks have been previously implemented in a side-by-side or 90° angle setting with a shared or separate workspaces for each subject (*Brosnan et al., 2017*; *Chang et al., 2013*; *Dal Monte et al., 2020*; *Falcone et al., 2017*; *Ferrari-Toniolo et al., 2019*; *Fujii et al., 2007*; *Haroush and Williams, 2015*), using back-to-back nontransparent monitors (*Jahng et al., 2017*; *Tang et al., 2016*), or in a table-like setting with opposing subjects acting in a horizontal workspace that is not in the line of sight between the subjects' faces (*Báez-Mendoza et al., 2016*; *Ong et al., 2021*; *Yoshida et al., 2011*). Only a few studies utilized transparent face-to-face arrangement of the subjects, using a video projector and two semitransparent mirrors virtually placing the stimuli into the shared plane (*Ballesta and Duhamel, 2015*), or incorporating physical targets into transparent Plexiglas screen (*Vaziri-Pashkam et al., 2017*). Similarly, with our design, we aimed for maximal availability of mutual social signaling in a face-to-face setting, a shared vertical workspace with computer-controlled stimuli in the direct line of sight, without risk of injuries due to physical contact, and suitability across species. We achieved this using a novel transparent display (1920 × 1080 pixels, 121 × 68 cm, 60 Hz, EYE-TOLED-5500, Eyevis, Reutlingen, Germany), amended for dual-side touch sensitivity (PQLabs, G5S, Freemont, CA, USA) with a custom-built 'sandwich' construction. Sitting on either side of the display, subjects saw each other and the same stimuli. Two proximity sensors (Carlo Gavazzi CA18-CA30CAF16NA, Lainate, Italy) per subject for the left and right hand, respectively, mounted below the display ('home' buttons) and the two touch panels mounted on either side of the display registered hand positions of both agents, at 240 Hz temporal resolution. Experimental control and stimulus presentation were implemented using the EventIDE software package (Okazolab, Delft, The Netherlands). Liquid reward for monkeys was delivered via computer-controlled peristaltic fluid pumps, for every correctly performed trial.

## Dyadic decision-making task

### Bach-or-Stravinsky game

Since we are interested in the effect of mutual action visibility on coordination behavior, we implemented a transparent version of a Bach-or-Stravinsky (BoS) game, in which each player's time-continuous visuomotor behavior can be seen by the other player. Conceptually, in the BoS game two agents are choosing between going to Bach or Stravinsky concerts. Agent A prefers Bach, agent B prefers Stravinsky; yet, both prefer going to the concert together (*Osborne and Rubinstein, 1994*).

Thus, agents wish to coordinate their behavior but have conflicting interests. This is a classic 2×2 non-zero-sum game (also known as the Battle of the Sexes) with two pure strategy Nash equilibria (Bach-Bach, or Stravinsky-Stravinsky), and one less efficient mixed strategy Nash equilibrium, where the agents go to their preferred event more often than the other, but the reward per player is below the reward of even the less rewarded player in the two pure Nash equilibria (*Kilgour and Fraser, 1988*; *Rapoport, 1967*).

In our implementation, the agents were choosing between two options represented by two differently colored targets placed on the left and the right side of the display. Each target color was associated with a higher reward for one of the agents resulting in individually preferred targets. This reward schedule, 2 reward units for one color and 1 unit for the other color, was trained in 'solo' trials to establish an 'own' color preference in each subject. In dyadic trials, an agent selecting own preferred target was assured to get at least 2 reward units, while selecting the other's target (individually non-preferred target) yielded at least 1 unit. Additionally, when both agents selected the same target, a bonus of 2 reward units was added to the payoff of each agent. Thus, the maximum average joint reward (3.5 units) was obtained on coordinated trials when one agent selected own preferred color (getting 4 units) while the other agent chose non-preferred color (getting 3 units). Note that on any given coordinated trial, i.e., when agents selected the same target, this payoff matrix resulted in an unequal (i.e., unfair) reward distribution (*Figure 1C*). Hence, the BoS paradigm probes both the ability to realize that coordinated target selection results in higher rewards than uncoordinated target selection, as well as the ability to perceive and counteract the unfairness/conflict situation.

## The task mechanics

On every trial, each subject chose between the individually-preferred and non-preferred target on the touchscreen. Targets were light blue circles of 35 mm diameter with either a red (individually-preferred target for agent A) or a yellow (individually-preferred target for agent B) rings (for better visualization, we replaced yellow and red rings with blue and red solid targets in the figures and the text).

Subjects had to place both hands on the two home buttons for 500 ms to start a trial after the inter-trial interval (ITI). This allowed us to control which effector (acting hand) was used (*Figure 1D*). Then an initial fixation target without a colored ring appeared on the display (10 cm below eye level). Subjects had 1500 ms to touch this initial fixation target with the instructed hand. After both subjects touched the fixation target they had to hold it for 500 ms. Then two choice targets appeared at 1 of 3 different pairs of positions (140 mm to the left and right of the central fixation target and either at the same height as the central fixation target or 35 mm below or 35 mm above it). We randomized the 'red' location equally over all six positions, balanced in sets of 18 trials with each of the six 'red' positions appearing three times, and vertically mirrored the opposite location. Simultaneously with targets' appearance the initial fixation target disappeared, serving as a go signal. Subjects had 1500 ms to make their choice and touch one of the targets. After both subjects acquired their chosen target, selected target(s) brightened, and subjects needed to hold the hand on the target for another 500 ms. At that point, the choices were evaluated and rewards were dispensed according to the payoff matrix. The amount of reward earned by each subject was signaled by two sequential series of auditory pulses (always first for the subject on side B then on side A), with a different pitch for each subject. Each pulse was constructed as a harmonic series with 12 overtones and a fundamental frequency of 443 Hz for side A and 733 Hz for side B to provide distinct sounds for each agent. For the monkeys, we delivered liquid as a reward (approximately 0.14 ml per pulse) concurrently with the auditory pulses directed towards the respective monkey's side; humans were instructed to expect 'a few' cents per pulse, and the accumulated earnings were paid out as a lump sum after the experiment. After the reward period, subjects had to wait for an inter-trial interval of 1500 ms before they could initiate the next trial.

The side of the targets was randomized on each trial, i.e., in ~50% the red target was on the right, the blue on the left, and vice versa. This design can lead to the three following coordination patterns (or any mix of the three):

1. Coordinating statically by both agents repeatedly selecting the same fixed color of the target, irrespective of unfair distribution of the rewards.
2. Coordinating statically by both agents repeatedly selecting the same fixed side. Due to color/side randomization, this pattern 'automatically' ensures a fair reward distribution.

3. Coordinating dynamically by both agents selecting the same target, while picking from both colors and both sides trial-by-trial (e.g. trial 1 red right, trial 2 red left, trial 3 blue left, … etc.). This could result in a fair or unfair reward distribution depending on the ratio of red to blue color selection.

Subjects were not informed about the randomization procedure, so recognizing the 'automatic' fairness of the 2nd pattern would have required keeping a count of color occurrences per side from both subjects.

## Human procedure

Human subjects were recruited via a university job website and pairs were selected based on matching schedules. Subjects were given a brief introduction to the experiment and a set of verbal instructions on the task mechanics (see Appendix 1, *Supplementary file 5*). This material included the information 'Your reward will depend on your own and your partner's choice' and a very basic description of the task ('You will have to choose one of the two circles presented to you. […] You will have to decide and respond quickly.'), but did not include details of the payoff matrix. Participants had to infer the task rules by exploration, similar to the macaques (*Brosnan et al., 2012*). After the joint introduction, each subject alone performed 100 individual (solo) trials, to learn how to operate the touchscreen and to develop a preference for one of the two color targets (in five pairs, we only used 50 solo trials to reduce the experiment's duration for logistical reasons, *Supplementary file 1*). All but one subject who was trained on 100 solo trials and converged to >75% own color selection did so already in the trials 26–50. Subjects were positioned ~50 cm from the display with the height of the chairs adjusted such that both subjects' eyes were ~121 cm above the ground. Subjects were instructed to use the right hand. After the solo training, both subjects entered the setup for the main dyadic task which lasted for 300 or 400 trials. Subjects were not informed about the number of trials in a session. Each session lasted approximately 1.5 hr. After the experiment, we conducted an individual debriefing (see Appendix 1, *Supplementary file 5*) and paid the earned reward separately to each subject (10.5–26 Euro, mean 19 Euro, *Supplementary file 1*).

## Macaque procedure

Macaques were brought to the set up in their individual primate chairs. The chairs were positioned such that the eyes were 30 cm from the display. The monkeys had previously been trained to perform the basic task structure (hands on proximity sensors, reach to the initial fixation target with the instructed hand, and select one of the presented choice targets by reaching to it). Dyads containing monkeys (monkey-monkey, and monkey-human confederate) were instructed to use the left hand. The animals performed the solo version of the task with differential rewards to develop a preference for one of the color targets and were only paired with a conspecific after selecting a higher rewarded target in ≥75% of the trials. Thereafter, pairs of macaques worked together in the dyadic version of the task, for 11 ± 7 sessions (range 4–25). After the dyadic task, we offered monkeys the opportunity to obtain additional liquid in the solo version of the task for as long as they continued to work (mean 169 trials, median 73, range 0–1163), essentially making participation in the dyadic trials optional.

## Data analysis

We computed the following six *aggregate measures of choice behavior and coordination,* explained below: fraction of choosing own color, fraction of choosing (objective) left side, mutual information for color choice, mutual information for side choice, average reward, and DCR. Unless noted otherwise, we report these measures for the last 200 trials of each session in order to assess the 'steady-state' behavior after allowing for an initial period of exploration.

## Fraction of choosing own/left

The FCO (fraction of choosing own) is the fraction of trials (within a certain window) where an agent has selected the individually preferred color (red for agent A and blue for agent B). Similarly, the FCL (fraction of choosing left) is a fraction of trials where an agent has selected the target on the 'objective' left side of the display (which is the left side for agent A and right side for agent B). Fractions range from 0 to 1. For figures showing FCO/FCL over the course of a session, we also calculated both

measures for the session as a whole, and in running windows of $w=8$ trials. In the latter case, FCO and FCL can take the values of 0, 1/8, 2/8, …, 1. For instance, FCO = 0 means that an agent has selected individually non-preferred target for eight trials in a row.

## Mutual information

Mutual information (MI) represents the reduction of uncertainty regarding the values of one time series provided by knowing the values of the other time series. Here, we consider mutual information for the color of the target (MI color) and side (MI side) choices of the two agents, showing how much information the target/side choices of one agent provide about the respective choices of the other. Mutual information is measured in bits. Since both target and side choices are binary (in each trial an agent selects either the preferred or non-preferred target and either the left or right side), both MI color and MI side range from 0 bit (the choices of one agent provide no information about the choices of the other) to 1 bit (the choices of one agent can be inferred precisely from the choices of the other agent).

For instance, if both agents select the objective left target in every odd trial and the objective right target in every even trial, both MI color = MI side = 1, since for every trial the target and the side selected by one agent entirely describe the target/side selected by the other. At the same time, the choices of each agent individually are highly uncertain: both sides and both targets are selected with the same probability of 0.5. If both agents constantly select the objective left side, this would result in MI color = 1 due to side-randomization of target color, but MI side = 0, since there is no uncertainty regarding the side selection and thus no additional knowledge about the other's choice can reduce the uncertainty.

Formally, mutual information of time series $X = (X_t)$ and $Y = (Y_t)$ is given by,

$$\text{MI}(X, Y) = \sum_{x,y} p(X_t = x, Y_t = y) \log_2 \frac{p(X_t = x, Y_t = y)}{p(X_t = x) p(Y_t = y)}$$

(1)

where $p(X_t = x)$ is the probability of the value $x$ in time series $X$, $p(Y_t = y)$ is the probability of the value $y$ in time series $Y$, and $p(X_t = x, Y_t = y)$ is the joint probability to simultaneously have values x and y in time series $X$ and $Y$, respectively, and x and y can be either 0 or 1, so that the sum is over all four combinations. Since in our case time series X and Y have finite length, we simply replace probabilities with relative frequencies. This is known as a naïve estimation of mutual information, but it is sufficiently precise for binary time series (**Grassberger, 1988**; **Steuer et al., 2002**).

To test whether the MI values were significantly different from zero, we generate Whittle surrogates for the given choice time series and estimate from them the threshold for the given significance level (p=0.01 in our case) (**Pethel and Hahs, 2014**).

## Average reward

Average reward (AR) is computed as the average of an agent's payoff across the session. Note that the reward of an individual agent can be in the range of 1–4 units, while the joint reward of a pair ([reward A+reward B]/2) cannot exceed 3.5 (since when one agent gets payoff of four, the other agent gets three). The average reward for completely random independent choices of two agents with 50% probability for either target is 2.5 (but note that an achieved reward of 2.5 is not a positive proof of independent choices).

## Dynamic coordination reward (DCR)

DCR is the surplus reward of the two agents compared to the reward they would get by playing randomly. By *playing randomly* we mean that the choices of the agents in each round are independent of the history and of the current choices of the partner. The 'random reward' is computed by selecting the two color targets with the same probabilities as actually observed in the two agents, but randomly permuting the choices over trials. For our payoff matrix, the range of DCR is [–1,1], with –1 corresponding to very inefficient playing (alternating selection of the two anti-coordination options), while 1 corresponds to very efficient playing with explicit coordination (for instance, turn-taking). DCR is hence a measure of dynamic (reciprocal) coordination. For instance, if both agents would coordinate

statically by constantly selecting one and the same side, this would result in DCR = 0, even though this coordination pattern still yields the maximum average reward of 3.5.

Formally, DCR is defined as the actual average reward of a pair ($R_{\text{actual}}$) minus the reward the agents would get if they were playing randomly ($R_{\text{PR}}$). DCR = $R_{\text{actual}} - R_{\text{PR}}$. The reward for playing randomly ($R_{\text{PR}}$) depends only on eight probabilities, four for each agent. Below index $i$ indicates the agent and stands for either A or B:

> $P_{i,1,\text{left}}$ - probability to select non-preferred objective left target (which is the left side for agent A and right side for agent B),
> $P_{i,1,\text{right}}$ - probability to select non-preferred objective right target,
> $P_{i,2,\text{left}}$ - probability to select preferred objective left target,
> $P_{i,2,\text{right}}$ - probability to select preferred objective right target.

Note that these probabilities are not independent on each other. First, for $i = $ A, B, it holds.

$$P_{i,1,\text{left}} + P_{i,2,\text{left}} + P_{i,1,\text{right}} + P_{i,2,\text{right}} = 1 \tag{2}$$

Second, it holds,

$$P_{\text{A},2,\text{left}} + P_{\text{A},1,\text{right}} = P_{\text{B},1,\text{left}} + P_{\text{B},2,\text{right}} = Q_{\text{left}} \tag{3}$$

and,

$$P_{\text{A},1,\text{left}} + P_{\text{A},2,\text{right}} = P_{\text{B},2,\text{left}} + P_{\text{B},1,\text{right}} = Q_{\text{right}} \tag{4}$$

where $Q_{\text{left}}$ and $Q_{\text{right}}$ are the probability of agent A's preferred target to appear on the left and on the right, respectively (and of B's preferred target appear on the right and on the left). Given the independent random selection from trial to trial described above, Q-values should approximate 0.5 for larger N of trials.

Average reward of two agents for playing randomly is computed as follows:

$$R_{\text{PR}} = 3.5\left(p_{4,3} + p_{3,4}\right) + 2p_{2,2} + p_{1,1} \tag{5}$$

where $p_{a,b}$ is the probability that by random playing agent A gets reward $a$ and agent B reward $b$. These probabilities are given by the following equations:

$$p_{1,1} = \frac{P_{\text{A},1,\text{right}} \cdot P_{\text{B},1,\text{left}}}{Q_{\text{left}}} + \frac{P_{\text{A},1,\text{left}} \cdot P_{\text{B},1,\text{right}}}{Q_{\text{right}}} \tag{6}$$

$$p_{2,2} = \frac{P_{\text{A},2,\text{left}} \cdot P_{\text{B},2,\text{right}}}{Q_{\text{left}}} + \frac{P_{\text{A},2,\text{right}} \cdot P_{\text{B},2,\text{left}}}{Q_{\text{right}}} \tag{7}$$

$$p_{4,3} = \frac{P_{\text{A},2,\text{left}} \cdot P_{\text{B},1,\text{left}}}{Q_{\text{left}}} + \frac{P_{\text{A},2,\text{right}} \cdot P_{\text{B},1,\text{right}}}{Q_{\text{right}}} \tag{8}$$

$$p_{3,4} = \frac{P_{\text{A},1,\text{right}} \cdot P_{\text{B},2,\text{right}}}{Q_{\text{left}}} + \frac{P_{\text{A},1,\text{left}} \cdot P_{\text{B},2,\text{left}}}{Q_{\text{right}}} \tag{9}$$

To see why this is the case, consider, for instance, $p_{1,1}$. Both agents get reward of 1 when they both select the other's preferred target, either when target of agent A appears on the left side (probability of this is encoded by the first term) or on the right side (second term).

To compute confidence intervals (CIs), we use the fact that $\text{FCL}_i$ and $\text{FCO}_i$ for an agent $i$ can be considered as binomially distributed, thus radiuses of their CIs $\Delta\text{FCL}_i$ and $\Delta\text{FCO}_i$ can be estimated by the classic method of approximating the distribution of error around binomially-distributed observation with a normal distribution. To obtain the CI for DCR, it is sufficient to compute maximal and minimal possible DCR given that FCO and SCL of the two agents are within the respective CIs. Note that $\text{FCL}_i = P_{i,1,\text{left}} + P_{i,2,\text{left}}$, $\text{FCO}_i = P_{i,2,\text{left}} + P_{i,2,\text{right}}$ , thus all the probabilities necessary for calculating DCR can be computed from $\text{FCL}_i$ and $\text{FCO}_i$ given that $Q_{\text{left}}$ and $Q_{\text{right}}$ are fixed to 0.5 in the reported experiments. Simple analysis reveals that minimum and maximum DCR should be at the edges of the 4-D CI formed by $\text{FCL}_i$ and $\text{FCO}_i$ of the two agents, which reduces the problem to testing 16 DCR values computed for $\text{FCL}_i \pm \Delta\text{FCL}_i$ and $\text{FCO}_i \pm \Delta\text{FCO}_i$ .

For the main analysis presented in the Results, we used last 200 trials to compute DCR values. Using the last 150 or 250 trials for DCR analysis resulted in the exact same 10 human pairs with significant results as using the last 200.

## Action time measurements

We measured the reaction time from the onset of the choice targets to the release time of the initial fixation target, $t_{\text{release}}$, and to the acquisition time of the selected target, $t_{\text{acquisition}}$, individually for each subject. We then calculated the movement time $t_{\text{movement}}$ and action time $t_{\text{action}}$ as follows:

$$t_{\text{movement}} = t_{\text{acquisition}} - t_{\text{release}} \tag{10}$$

$$t_{\text{action}} = t_{\text{release}} + \frac{t_{\text{movement}}}{2} \tag{11}$$

We use $t_{\text{action}}$, the halfway point between $t_{\text{release}}$ and $t_{\text{acquisition}}$, i.e., the duration from target stimulus onset to the half-time of the reach movement, as a proxy for the estimated time at which the trajectory of each subject's reach movement should be evident for the other agent. Typical movement time values (mean ± SD across trials) in our experiment ranged from 314 ±104 ms for humans, 171 ± 77 ms for macaques, and 180 ± 64 ms for the confederate-trained macaque pair.

## Action time difference analysis

To compare action times (AT) between agents we performed three tests. First, we compared the action times over all trials between both agents of a pair to assess stable differences that might indicate temporal leader-follower relations. Second, we compared per-trial difference in action times of the two agents (AT A – AT B) between coordinated trials on A's preferred color and coordinated trials on B's preferred color. Third, we compared the absolute per-trial difference in action times of the two agents between coordinated and uncoordinated trials. The absolute per-trial difference is equivalent to subtracting for each individual trial the action time of the faster agent (in this trial) from the action time of the slower partner. All comparisons used a two-sample t-test using Satterthwaite's approximation to allow for unequal variance.

## Estimated probability to see partner's choice

When an agent is acting slower than the partner, there is a chance for this agent to see the partner's choice and use this information to make the own choice. We therefore modeled the probability to see the partner's choice as a logistic function of the difference of the agents' action times, in each trial. The logistic function $p_{\text{see}}(t)$ had an inflection point at 50 ms ($p_{\text{see}}(50) = 0.5$) and reached its plateau phase at 150 ms ($p_{\text{see}}(150) = 0.98$). Formally, the function was given by the following equation

$$p_{\text{see}}(t) = \frac{1}{1 + e^{-k(t - \Delta T_0)}} \tag{12}$$

with $k = 0.04$ (steepness of the slope) and $\Delta T_0 = 50$ ms (inflection point). The values of $p_{\text{see}}$ were used for the analysis of correlation with probability of selecting the other's color (*Figure 6*). We also tested a wide range of these parameters ($k$ fixed at 0.04, $\Delta T_0 = \{12.5, 25, 50, 75, 100, 200\}$ ms and $\Delta T_0$ fixed at 50 ms, $k = \{0.01, 0.02, 0.04, 0.08, 0.16\}$), and confirmed that the values of correlation were robust with respect to these parameters; only the longest $\Delta T_0 = 200$ ms resulted in a noticeable drop in resultant correlations.

## Acknowledgements

We thank Elisheba Crecca, Tarana Nigam, and Roberta Nocerino for their help with the data collection, and Daniela Lazzarini, Janine Kuntze, Sina Plümer and Klaus Heisig for technical support. This work was supported by a funding from the Ministry for Science and Education of Lower Saxony ("Top Level Research in Lower Saxony") and the Volkswagen Foundation through the "Niedersächsisches Vorab", https://www.volkswagenstiftung.de/en/funding/niedersaechsisches-vorab (JF, AG, ST, IK). Additional support was provided by the Leibniz Association through funding for the Leibniz Science-Campus Primate Cognition, https://www.primate-cognition.eu (JF, AG, ST, IK), the Leibniz Collaborative Excellence grant K265/2019 Neurophysiological mechanisms of primate interactions in dynamic

sensorimotor settings (AG, ST, IK) and by the German Research Foundation (Deutsche Forschungs-gemeinschaft, DFG), SFB 1528 – Cognition of Interaction (http://www.sfb1528.uni-goettingen.de), project Z01 (AG, IK). The funders had no role in study design, data collection and analysis, decision to publish, or preparation of the manuscript.

## Additional information

### Funding

| Funder | Grant reference number | Author |
|---|---|---|
| Lower Saxony Ministry of Science and Culture | Niedersächsisches Vorab | Julia Fischer Alexander Gail Stefan Treue Igor Kagan |
| Leibniz Association | Leibniz ScienceCampus Primate Cognition | Julia Fischer Alexander Gail Stefan Treue Igor Kagan |
| Leibniz Association | Leibniz Collaborative Excellence grant "Neurophysiological mechanisms of primate interactions in dynamic sensorimotor settings", K265/2019 | Alexander Gail Stefan Treue Igor Kagan |
| Deutsche Forschungsgemeinschaft | SFB 1528 Cognition of Interaction, project Z01 | Alexander Gail Igor Kagan |

The funders had no role in study design, data collection and interpretation, or the decision to submit the work for publication.

### Author contributions

Sebastian Moeller, Conceptualization, Resources, Data curation, Software, Formal analysis, Validation, Investigation, Visualization, Methodology, Writing – original draft, Writing – review and editing; Anton M Unakafov, Conceptualization, Resources, Software, Formal analysis, Validation, Investigation, Visualization, Methodology, Writing – original draft; Julia Fischer, Funding acquisition, Writing – review and editing; Alexander Gail, Stefan Treue, Conceptualization, Supervision, Funding acquisition, Project administration, Writing – review and editing; Igor Kagan, Conceptualization, Data curation, Supervision, Funding acquisition, Visualization, Writing – original draft, Project administration, Writing – review and editing

### Author ORCIDs

Sebastian Moeller http://orcid.org/0000-0002-0381-6449
Julia Fischer http://orcid.org/0000-0002-5807-0074
Alexander Gail http://orcid.org/0000-0002-1165-4646
Igor Kagan http://orcid.org/0000-0002-1814-4200

### Ethics

Human subjects: Experiments were performed in accordance with institutional guidelines for experiments with humans and adhered to the principles of the Declaration of Helsinki. The experimental protocol was approved by the ethics committee of the Georg-Elias-Mueller-Institute for Psychology, University of Gottingen (GEMI 17-06-06 171).

Research with nonhuman primates represents a small but indispensable component of neuroscience research. The scientists in this study are committed to the responsibility they have in ensuring the best possible science with the least possible harm to the animals (Roelfsema and Treue, 2014). The experimental procedures were approved by the responsible regional government office (Niedersaechsisches Landesamt fuer Verbraucherschutz und Lebensmittelsicherheit (LAVES), permits 3392-42502-04-13/1100 and 3319-42502-04-18/2823), and were conducted in accordance with

the European Directive 2010/63/EU, the corresponding German law governing animal welfare, and German Primate Center institutional guidelines.

## Decision letter and Author response
Decision letter https://doi.org/10.7554/eLife.81641.sa1
Author response https://doi.org/10.7554/eLife.81641.sa2

---

## Additional files

### Supplementary files
• Supplementary file 1. Table S1: human pairs.

• Supplementary file 2. Table S2: sequence of macaque pairings and confederate training sorted by the start date, and action times.

• Supplementary file 3. Table S3: action time correlations in macaque pair FC.

• Supplementary file 4. Table S4: action time correlations in humans.

• Supplementary file 5. Appendix 1: Supplementary materials and methods.

• MDAR checklist

### Data availability
The datasets used in the current study and the links to the public GitHub code repositories are uploaded to a public Open Science Framework data repository (https://osf.io/f5u8z/).

The following dataset was generated:

| Author(s) | Year | Dataset title | Dataset URL | Database and Identifier |
|---|---|---|---|---|
| Moeller S, Kagan I | 2023 | Human and macaque pairs employ different coordination strategies in a transparent decision game | https://osf.io/f5u8z/ | Open Science Framework, 10.17605/OSF.IO/F5U8Z |

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

# Appendix 1

## Supplementary materials and methods
### Written instructions read by human subjects
### Description of the study
This study is the investigation of the behavioral correlates of social decision-making while playing a game with a partner. Every day we have to make decisions that depend not only on our own needs and goals but also on the needs and goals of others: for instance, while working on a project with colleagues, planning vacation with friends and family, getting on the bus or shopping groceries, etc. With the help of the task presented in this study, we will investigate how people make such decisions.

### The course of the study
You will complete one session of a decision-making task on the computer together with your partner. You will have to choose one of the two circles presented to you. Your partner will have to perform the same task on his/her side of the touchscreen. You will have to decide and respond quickly. If either you or your partner is too slow, the trial will be aborted without any reward. Your reward will depend on your own and your partner's choice. After the decision is made, you both will receive different auditory feedback, denoting your reward and the reward of your partner. Please do not talk to your partner during the session. After the session, we will ask you several questions about the experiment.

### Instructions read to human subjects by experimenter

1. You rest both hands on the gray board at the two round objects (touch sensors).
2. A central touch target will appear.
3. Move your right hand to the target and hold. Use the right hand during the entire session.
4. While the target brightens up keep holding your finger on the target.
5. Then two colored choice targets appear and the central target disappears.
6. Make your choice and touch the chosen target within 1.5 s.
7. All touched targets will brighten up, keep holding until the targets disappear.
8. Please note that both selected targets will brighten up: in case both players selected the same target only that single target will brighten; in case both players selected different targets, both targets will brighten.
9. Now, two streams of auditory beeps will signify the earned reward for each player: each beep corresponds to a few cents. Please try to learn the sound related to your reward during the training trials.
10. While the audio plays move the hand back to the two touch sensors.
11. Go to 1.

### Debriefing questions

1. Was there any recognizable system between the choice item and reward size?
2. How did you make your choice of which circles to choose?
3. How do you think the other player decides which circle to choose?
4. Do you think there is an optimal strategy, if so, what?
5. Did the possibility to sit in front of the partner have any impact on your choices?
6. Rate the cooperativity of your partner, with 1 fully cooperative and 7 fully competitive.
7. Rate the cooperativity of yourself, with 1 fully cooperative and 7 fully competitive.
8. How did you recognize the other player's choice?
9. Did you observe any relation between colors and reward magnitude per trial?

