## [Editor Report]

This study investigates and compares spontaneous turn-taking behavior in pairs of macaque monkeys and human participants. The study is well-designed and uses a novel format for dynamic interaction. The analyses are rigorous and support the overall conclusion that there are differences between species in their tendencies toward cooperative, mutually beneficial behaviors, with humans exhibiting more prosocial tendencies. This finding, as well as the rich description of pair interactions in each species, is likely to be relevant to a broad range of researchers interested in social behavior.

---

## [Decision Letter]

**Decision letter after peer review:**

Thank you for submitting your article "Human and macaque pairs employ different coordination strategies in a transparent decision game" for consideration by *eLife*. Your article has been reviewed by 3 peer reviewers, one of whom is a member of our Board of Reviewing Editors, and the evaluation has been overseen by Joshua Gold as the Senior Editor. The following individual involved in review of your submission has agreed to reveal their identity: Steve W C Chang (Reviewer #2).

The reviewers have discussed their reviews with one another, and the Reviewing Editor has drafted this to help you prepare a revised submission. Full comments by each of the reviewers are appended to this summary for your reference.

Essential revisions:

1. Some human pairs (about half) did not employ a turn-taking strategy. The authors should discuss potential reasons for these differences, for instance, whether some subjects were not as socially engaged or were less flexible when choosing different options. It would be most helpful if the authors are able to perform additional data analyses to characterize the differences between the two groups of humans (social attentional differences, reaction time differences, etc.) to examine why the turn-taking and non-turn-taking pairs differed.

2. The authors should further consider the degree to which maximizing one's own rewards could drive different strategies in the monkeys. Although cooperating maximized the group's rewards, it wasn't clear whether one monkey could do even better by behaving competitively.

3. It was felt that the manuscript would benefit from including additional information on the monkeys' interactions with the human confederate. Specifically, how did their behavior differ with humans compared to other monkeys, how long do effects of training last, and would the behavior of a trained monkey be expected to translate to an untrained monkey?

4. A notable limitation of the study's ability to compare across species is that the rewards given to the monkeys and humans differed qualitatively (primary vs. secondary rewards). Although this is touched on in the discussion, it was felt that further consideration is warranted. For instance, to what extent might motivation to obtain each outcome drive the tendency to be 'selfish' versus cooperative? Is there any evidence that monkeys would adopt different strategies if they were more sated?

5. Can the authors more precisely consider the role of action visibility? Specifically, whether visibility influences social cognition or aides sensorimotor coordination.

6. Finally, can the authors provide further discussion of the effects of learning and socialization on this task across species?

*Reviewer #1 (Recommendations for the authors):*

1. It would be helpful to know more about how the goal of maximizing reward might impact the behaviors described here. To start, how does turn-taking compare with the simpler strategies used by the monkeys in terms of reward received? Figure 2C seems to show that the average reward earned by two subjects was actually quite similar between monkeys and humans. The highest-earning human pairs (presumably those who were turn-taking) might have earned slightly more than the monkeys, but there doesn't appear to be a big advantage to turn-taking, in terms of simple reward maximization. Is this true, and if so, is it possible to interpret their behavior as arising from other advantages (e.g., placing value on a partner's reward specifically vs. perceived equity)? What about rewards received by each individual and not the average across pairs? If an agent only values rewards received by themself, is there a more or less optimal approach to this task?

2. Related to the above, it is notable that rewards given to the monkeys and humans differed qualitatively, as did their motivation to pursue them. The humans were receiving small amounts of money in a fairly abstract arrangement in which money was obtained at the end of the session, whereas the monkeys were thirsty and received fluid rewards continuously depending on trial outcomes. To what extent is the motivation to be 'selfish' versus cooperative driven by the motivation to obtain each outcome? For instance, if monkeys were not water deprived, would they be more likely to cooperate rather than compete? Is there any evidence that motivation to get the juice reward changed toward the end of the session?

3. Are there action time plots similar to those shown for the monkeys for the human pairs? Although Figure 6B is compelling, if I understand it correctly, it just shows that the humans' choices can't be predicted based on who responded first. But it's hard to understand, then, how this type of coordination would emerge if one participant didn't watch and follow the other. The results in 6B could occur if the person who responds faster also drives the shifts to the other's target from time to time, as suggested in Figure 7. Therefore, is it possible that there is a leader and follower in the human pairs (perhaps these roles switch off), but the leader just doesn't choose their own color reliably?

*Reviewer #2 (Recommendations for the authors):*

This manuscript is nicely organized and very thorough. However, it could be improved even further if some points can be discussed in more detail or with additional data analyses.

1. Related to Figure 2D: the authors mentioned that some human pairs (about half) were not employing the turn-taking strategy. From this figure, these pairs were actually not a small number (9 out of 19). The authors should discuss what might be the reason behind this. Is it possible that these subjects were not very socially engaged (i.e., not paying attention to others) or were less flexible when choosing different options (even in non-social situations)? It will be better if the authors could perform additional data analyses to characterize the differences between the two groups of humans (social attentional differences, reaction time differences, etc) to examine why they were different from the turn-taking pairs and more similar to the monkey pairs.

2. Related to Figure 5: It is still a little puzzling that monkeys' behaviors can be improved after being trained with humans. What was the reason for this change? Did the human subjects provide extra social cues to the monkeys in order to encourage them to pay more attention, or did the monkeys learn turn-taking simply from the diverse choice patterns of the human subjects? The authors need to have further discussion about this point.

3. It is not very clear whether the action visibility is influencing social cognition or simply aiding sensorimotor coordination. The authors should discuss this further in Discussion.

---

## [Author Response]

Essential revisions:1. Some human pairs (about half) did not employ a turn-taking strategy. The authors should discuss potential reasons for these differences, for instance, whether some subjects were not as socially engaged or were less flexible when choosing different options. It would be most helpful if the authors are able to perform additional data analyses to characterize the differences between the two groups of humans (social attentional differences, reaction time differences, etc.) to examine why the turn-taking and non-turn-taking pairs differed.

This point is related to the questions raised by Reviewer 1, Recommendations to the authors Comment 3 [R1, C3] and Reviewer 2, Recommendations to the authors Comment 1 [R2, C1].

We thank the Reviewers for bringing up this question. First, we performed additional analysis of action times as well as absolute action time differences between the two agents – i.e. mean(|ATA – ATB|) – to compare the two groups: turn-takers and non turn-takers. There was no significant difference in mean action times (t-test correcting for unequal variance: turn-takers mean (M, in ms) 577.3 SD 80.7 N 20; non turn-takers M 536.1 SD 62.9 N 18; t(35.3) 1.766; p 0.086). Therefore, both groups appear to have a similar engagement in the task. However, when comparing means of the absolute trial-by-trial action time differences between the two agents, turn-taking dyads had a larger difference than non turn-taking dyads. We added this information to Results (lines 416-422) and the revised Table S1 (Supplementary file 1) (columns 11-17):

“Furthermore, turn-taking dyads had a larger absolute action time difference (102.7 ± 38.7 ms, N 10) than non turn-taking dyads (61.3 ± 21.6 ms, N 9; t-test correcting for unequal variance t(14.4) 2.918; p = 0.011); this difference was mostly driven by the coordinated trials (105.0 ± 41.5, N 10 vs. 64.7 ± 24.1, N 7; t(14.7) 2.523; p = 0.024), while a similar trend in trials without coordination did not reach significance (93.8 ± 29.2, N 10 vs. 69.5 ± 39.7, N 9; t(14.6) 1.512; p = 0.152). This implies that turn-takers tended to pay more attention to partner’s actions and wait more for each other during the coordination.”

We emphasize however that the leader-follower dynamics also transpired in non-turn-taking dyads and in macaques statically coordinating on a fixed color or side. We now address this in lines 254-263:

“We next asked whether a difference in action times between the two agents in a pair influenced the coordination by affecting visibility of each other’s actions. Remarkably, all 5 macaque pairs coordinating on the fixed color converged on the faster monkey’s preferred color (Table S2, Supplementary file 2). Likewise, for the static side coordination, all 3 (out of 4) pairs where there was a significant difference in action times between the two agents converged on the faster monkey’s less effortful side closer to the acting arm (Table S2, Supplementary file 2). In humans, such static Leader-Follower relationship also transpired in non-turn-taking dyads coordinating on the fixed color (Table S1, Supplementary file 1), where the faster agent led to own color and the slower agent followed (e.g. pairs 10 and 14), or the faster agent led to the partner’s color and the partner followed (e.g. pair 18). This implies that the static coordination in macaques and some human dyads was shaped by the faster agent’s preferences.”

If the mutual action visibility serves as means of coordination, one may expect that the action time differences should be larger when coordination took place. Therefore, we also directly compared absolute AT differences between the two agents in coordinated vs non-coordinated trials (revised Table S1, Supplementary file 1, columns 18-26). Six of 10 turn-taker dyads showed a significant difference between coordinated and uncoordinated trials, as well as 3 of 9 non turn-taker dyads (the difference in ratios was not significant, Fisher’s exact test, p 0.3698). In 4 turn-taking dyads but also in 2 non turn-taking dyads the difference was larger for coordinated trials, further suggesting leader-follower dynamics that was not limited to turn-takers.

Since our human subjects, like macaques, learned the payoff contingencies while playing the game, we asked several debriefing questions after the experiment (see Supplemental Information). We extracted from these free form reports whether subjects explicitly understood that coordination on “same” targets was beneficial and whether they also realized that the two coordination options were more favorable for one or another player. We now discuss these additional insights in lines 145-148:

“Based on the debriefing responses (Supplementary Information), most subjects from the turn-taking dyads (17/20, 85%) realized the coordination benefit, as compared to only half of non turn-takers (9/18, 50%; Fisher’s exact test p=0.035). In addition, 14 of 20 turn-takers (70%) understood the payoff structure, compared to only 1 of 18 non turn-takers (6%; Fisher’s exact test p=0.00005).”

and lines 501-508:

“What sets the turn-takers apart from the dyads that coordinated on a fixed color, or did not coordinate well? Based on their debriefing responses, turn-takers displayed a much better understanding of the implicit reward rules such as the value of coordination and unfairness. The non turn-taker group was more diverse: some subjects did not understand the game or only understood some aspects; other subjects had a better understanding but were selecting selfishly or could only follow the partner. Furthermore, the turn-taking dyads utilized action visibility driven by action time differences more extensively, although some non-turn-taking dyads overall also exploited action visibility to coordinate on a fixed color.”

We also note that 4 subjects from non-turn-taking pairs (and no subjects from turn-taking pairs) showed <25% own target color selection in the end of solo training (revised Table S1, red font, now mentioned in the legend), in line with the Reviewer’s suggestion of less flexibility in non-social situations.

Finally, the Reviewer 1 is absolutely correct that in many human dyads the faster leaders also drove the shifts to the other's target, as highlighted in Figures 7 and S8, and lines 422-426:

“Hence, similarly to macaques, seeing the faster agent’s choice can help the slower agent to coordinate. But unlike the confederate-trained macaque pair, dynamic human coordination was not driven by the faster agents striving to select only the own preferred color (except human pair 12 shown in Figure 6B): for instance, in the turn-taking pairs 3 and 17, the faster agent B led both to own and A’s preferred colors, while A followed (Table S1, Supplementary file 1).”

To sum up, as a group human turn-takers explored more, better understood the game and used action visibility more to coordinate, but the leaders did not exclusively strive to select their preferred option – either due to understanding of fairness or because they did not realize the payoff asymmetry. The non turn-taker group was diverse, where some subjects did not understand the game or only understood some aspects, whereas other subjects had a better understanding but could only follow the partner. We also note that the proportion of turn-takers in our study, in absence of explicit payoff instructions, is similar to what has been reported in the literature in similar tasks with more explicit instructions, suggesting the facilitation by direct action visibility (lines 497-500):

“The observed turn-taking frequency was close to other variants of BoS games with explicit instructions of the payoff matrix, where approximately 50-60% of human subjects developed turn-taking (Arifovic and Ledyard, 2018; Ioannou and Romero, 2014; Melis et al., 2016; Sonsino and Sirota, 2003).”

While the number of human dyads in our study is not sufficient to make a systematic link between in-game behavior to other measures such as personality traits, this should be addressed in future work in a larger sample (lines 692-696):

“Finally, it would be very interesting to relate diverse in-game behavioral strategies observed in the human pairs to individual differences in cognitive and socio-economic personality traits (Rollwage et al., 2020; Zhao and Smillie, 2015), and link them to stable behavioral phenotypes (Peysakhovich et al., 2014; Poncela-Casasnovas et al., 2016). This however will require a much larger sample and personality-oriented study design.”

2. The authors should further consider the degree to which maximizing one's own rewards could drive different strategies in the monkeys. Although cooperating maximized the group's rewards, it wasn't clear whether one monkey could do even better by behaving competitively.

This point is related to the question raised by Reviewer 1, Recommendations to the authors Comment 1 [R1, C1].

It is an important point to clarify. To recap, the BoS paradigm is constructed such that coordination (selecting the same target) is invariably more rewarded than not coordinating (selecting different targets). This is related to the fact that converging on either of the two targets results in a game theoretic Nash equilibrium – a strategy in which no individual player can improve its payoff by changing course unilaterally. (For completeness, BoS has a third mixed Nash equilibrium in which players select the two choices with specific probabilities, but it is inefficient since the average reward per player is below the reward of even less rewarded player in the two pure – unfair – Nash equilibria). If an agent is dissatisfied with the current mode of coordination (e.g. always following partner’s color and getting only 3 units, or engaging in side-based on dynamic turn-taking and getting only 3.5 units, on average), he/she can try to force/nudge the partner to coordinate on his/her own preferred target by insisting on it, but for this to be a successful reward maximizing strategy the partner needs to follow suit – otherwise the agent initiating the change risks a reward loss if the partner does not follow. In other words (lines 118-124):

“In the dyadic trials, selecting the same target resulted in an additional reward on top of the individually trained values – such that choosing the same target was always better than choosing different targets, for both agents individually, and as a group. But in such coordinated trials, the agent whose individually preferred (“own”) color was selected receives a larger reward. Therefore, agents may attempt to behave selfishly/competitively in order to gain more by insisting on their own target color – which pays off as long as the partner accommodates.”

We indeed observed such behavior in both humans and macaques.

Concerning human dyads, as can be seen in the revised Figure 2C (where we now indicate turn-taking dyads), all 10 turn-taking dyads earned >3 units average joint reward, as compared to only 4 of the 9 non turn-taker dyads (this difference was significant; Fisher’s exact test p < 0.011). But as should be clearer now from the revised figure and lines 150-153:

“Note that turn-taking is not maximizing average joint reward over coordination on a fixed color (in fact, out of the three pairs reaching maximum average joint reward 3.5, only one did so by perfect turn-taking, while the other two used static coordination); it does however equalize the rewards within a pair.”

The turn-taking does optimize “fairness” (with both agents ideally getting 3.5 on average, rather than one getting 4 and other 3). For coordinating non turn-takers, the individual reward reaches maximum (4) for the selfish agent and 3 for the accommodating agent, lower than 3.5, but still higher than 2 for no coordination (e.g. dyads 10 and 18). The revised Figure 2—figure supplement 2B shows the individual rewards and now also indicates the turn-taking pairs.

For macaques, the most parsimonious explanation is that they indeed learned to maximize their own reward, either by finding one of the pure Nash equilibria (unfair to one of the two agents, but still better that no coordination), or by converging on the same side (which incidentally ensured fair distribution – we do not think that monkeys were motivated by fairness, see lines 669-672):

“Secondly, due to the task target color/side randomization, selecting targets on the same side of the display resulted in low effort, efficient and fair coordination. We cannot say if fairness was a contributing factor in convergence of the four macaque pairs to this strategy, although we deem it highly unlikely.”

As can be seen in revised Figure 2—figure supplement 2G which now indicates side strategy pairs, the individual reward is equalized in those macaque pairs but differs by ~0.7-1 units in fixed color-coordinating pairs.

To sum up, “well-coordinating” human dyads (all turn-takers and few non turn-takers) and all macaque dyads maximized their individual reward, achieving a high average reward above 3 units. The slight difference between well coordinating humans and macaques was not significant (Wilcoxon rank sum test: human dyads median 3.44 N 14; macaque dyads median 3.29 N 9, rank sum 192.5, p=0.297; we added this information to Figure 2 legend) and is likely due to more coordination lapses in macaques. For human and macaque dyads who coordinated statically on a fixed color, the difference between higher and lower individual rewards in a dyad was 0.7-1 units, but the accommodating agents still received more than they would for not coordinating and only selecting their own color.

3. It was felt that the manuscript would benefit from including additional information on the monkeys' interactions with the human confederate. Specifically, how did their behavior differ with humans compared to other monkeys, how long do effects of training last, and would the behavior of a trained monkey be expected to translate to an untrained monkey?

This point is related to the question raised by Reviewer 2, Recommendations to the authors Comment 1 [R2, C1]. We answer each of these important aspects below.

How did their behavior differ with humans compared to other monkeys?

The Reviewer is asking which aspects of macaque experience with the human confederate could have led to a turn-taking behavior in monkeys. As we now clarify in lines 280-283 and new Table S2:

“Before confederate training, both monkeys adopted either fixed side or fixed color strategy with the macaque partners (Table S2, Supplementary file 2). When paired with the confederate, both monkeys started with a strong bias for selecting their own preferred color but over time changed to reliable coordination (cf. Figure 4A with 4B, and Figure 4—figure supplement 1A with Figure 4—figure supplement 1B for another monkey).”

The human confederates did not intentionally provide additional social cues. The monkeys learned – in case of monkey F, fairly slowly, in the span of 15 sessions (Figure 4—figure supplement 2A-F) – to follow the diverse choices of consistent blockwise human turn-taker in order to maximize their reward. This learning was facilitated by the direct visibility of human actions.

While our new analysis shows that 8 of 9 monkey pairs, including pairs with monkeys F and C, converged on the faster agent’s preferred choice, we speculate that this type of “static” following was indeed different from trial-by-trial monitoring and following the dynamic human actions. We clarify these points in lines 265-279:

“The absence of dynamic coordination in macaque pairs does not preclude the possibility that a slower agent could have monitored the faster partner trial-by-trial in order to coordinate. Alternatively, the faster agent might have reduced the possible outcomes for the slower agent to two options by consistently selecting a fixed side or color, and the slower agent then found the best of the remaining options without resorting to the trial-by-trial following. To test if macaques can actively monitor and adjust to dynamic choice behavior, we trained two monkeys in the dyadic condition with a human confederate who followed a strict pattern of alternating between the two colors in blocks of approximately 20 trials (“confederate training”). If the rhesus monkeys were insensitive to partner’s diverse choices, the pattern of confederate’s choices should not affect monkey’s behavior.”

in lines 512-518:

“Although we found that all macaque pairs that exhibited a significant difference in action times between the agents (8 of 9 pairs) converged on the faster agent’s preferred choice, it seems that initially macaques did not rely on trial-by-trial action visibility. This is suggested by the fact that it required relatively long training before macaques started to follow a human confederate dynamically. Instead, we speculate that slower macaques adapted to their partner’s static choice by maximizing their reward over the remaining options.”

and lines 614-618:

“Very different dynamics transpired in additional experiments with macaques. Pairing them with a human confederate who consistently alternated between colors in short blocks but did not intentionally provide any additional social cues demonstrated that macaques are capable to closely follow partner’s actions (Nougaret et al., 2019), and adopt the imposed “turn-taking” to maximize their reward.”

How long do effects of training last?

As shown in Author response image 1, the confederate-following behavior, once established, was persistent across multi-week/multi-month breaks in confederate training. We did not explore this question specifically in our study and hence only can present data from the circumstantial breaks that happened in the course of continued training in the context of another study.

**Author response image 1. sa2fig1:** Confederate-following behavior was preserved across training breaks. In both monkey C and E following behavior – as measured by the dynamic coordination reward (DCR) – directly before and after a training break with the confederate stayed unchanged. The numbers indicate break duration in days. We note that each post-break training started with multiple sessions in the solo condition to re-establish the monkey’s color preference and to get him accustomed to the task, however following behavior was established essentially immediately in the first post-gap confederate session after each of the breaks.

Would the behavior of a trained monkey be expected to translate to an untrained monkey?

We thank the Reviewer for this insightful question. It seems that for turn-taking behavior to occur a single confederate-trained monkey is not sufficient, since turn-taking is a joint strategy that requires active participation of both agents. We now mention this in the Discussion (lines 634-639) and in the legend of the new Table S2:

“Notably, we considered the interactions in the pairs where one monkey underwent the confederate training, and other did not (Table S2, Supplementary file 2). In all 4 instances, static strategies prevailed, suggesting that turn-taking as a joint strategy behavior requires active participation of both agents. We note, however, that in all 3 cases of fixed color strategy, the confederate-trained monkeys accommodated the selfish choice of their partners, perhaps because they had learned to monitor and follow partner’s actions.”

4. A notable limitation of the study's ability to compare across species is that the rewards given to the monkeys and humans differed qualitatively (primary vs. secondary rewards). Although this is touched on in the discussion, it was felt that further consideration is warranted. For instance, to what extent might motivation to obtain each outcome drive the tendency to be 'selfish' versus cooperative? Is there any evidence that monkeys would adopt different strategies if they were more sated?

This point is related to the question raised by Reviewer 1, Recommendations to the authors Comment 2 [R1, C2].

It is indeed an important point; we agree that the difference between primary vs secondary rewards might be crucial and it is essential to consider potential changes in motivation during macaque sessions. We considered the two aspects of this question: (i) coordination vs no coordination and (ii) selfish vs benevolent coordination.

Concerning the first aspect, in the BoS game the coordination is invariably better than no coordination (both for an individual and a dyad, please see also the point 2 above). Therefore, “selfish” tendency to select only own preferred color would result in less reward, unless a partner always accommodates. A “rational” less sated monkey should select the coordination on either color to gain more reward, so from this perspective one might expect that there would be more coordination at the start of the sessions compared to the end. We however observed the effect in the opposite direction: when the difference was significant, the coordination (as assessed by the average joint reward) was stronger in the last 200 trials of a session compared to the first 100. In fact, we see a similar effect in the human dyads, suggesting the development of coordination irrespective of the reward type (see Figure 2—figure supplement 1A,C,D). We now illustrate how the individual average reward, average action time, and the control measure “fraction of choosing left” (pertinent for the side coordination strategy) change between the start and the end of a session (Author response image 2 and 3). In line with the average joint reward, the individual rewards increased significantly in early macaque and human sessions (Author response image 2,C), and across all sessions in confederate-monkey pairs, significantly for monkey F and marginally for monkey C (Author response image 3,B).

**Author response image 2. sa2fig2:** Changes in reward, side bias, and action time between start and end of sessions for different groups. All panels show data for each individual subject in a pair (agent A in red, agent B in blue); the x-axis shows data from the first 100 trials, the y-axis data from the last 100 trials; rows show different groups; columns show the average reward, the fraction of choosing left, and the average action time. The numbers on top of each plot show the results of Wilcoxon signed rank tests on whether the value changed between the early and the late trials.

**Author response image 3. sa2fig3:** Changes in reward, side bias, and action time between start and end of sessions for different groups. All panels show data for each individual subject in a pair (agent A in red, agent B in blue); the x-axis shows data from the first 100 trials, the y-axis data from the last 100 trials; rows show different groups; columns show the average reward, the fraction of choosing left, and the average action time. The numbers on top of each plot show the results of Wilcoxon signed rank tests on whether the value changed between the early and the late trials.

Of course, we cannot rule out that in a less sated monkey the urge of (reflexively) selecting “own” pre-trained color might override the prospect of the coordination benefit, resulting in less coordination at the start of a session. But we cannot disentangle this effect from a need to (re)establish the coordination regime. We emphasize however that both prior to and during the confederate training, there was no significant difference in the coordination strength or individual rewards between the start and the end, in the late sessions – on which we base our strategy descriptions (while this difference was present in early sessions when the dyadic learning just started). In other words, the selection behavior was fairly stable in late macaque-macaque and last confederate-macaque sessions. Likewise, there was no significant increase in rewards in the macaque FC pair after the confederate training.

No change in the fraction of choosing left confirms that there was no shift in side-based strategies in the course of sessions. Finally, we checked how the action times changed in the course of each session. There was an increase from early to late trials (except in confederate-trained FC pair and in humans). This increase can be due to decreased motivation because of satiation and/or fatigue. But to re-iterate, these effects did not result in a shift of strategy.

Concerning the second aspect, selfish vs benevolent coordination, we now analyzed if the proportion of selfish vs benevolent coordinated trials might change in the course of each session (Author response image 4). In the macaque dyadic late sessions, the behavior was very stable, without any change. Likewise, in the last 4 confederate-training sessions of each monkey (where both monkeys successfully followed the confederate), there was no significant change in the course of any session. Only in the dynamically-coordinating macaque FC pair after the confederate training, in several sessions one of the two agents became more selfishly insistent on his preferred color towards the end of the session – while the other accommodated. We now added a note of this effect in the legend to Figure 5—figure supplement 1:

**Author response image 4. sa2fig4:** Fraction of coordination on B’s preferred color, in early and late trials. Note that “fraction coordinating on B’s color” is complementary to “fraction coordinating on A’s color”; here we present data from B’s perspective because macaques were B agents when training with the confederate. Changes between early and late trials are visible as deviation from the unity diagonal. Individual sessions that exhibited a significant change are shown by filled symbols (Fisher’s exact test on counts of coordinating on B’s color vs A’s color with Bonferroni correction for the number of tested sessions). Locations in the upper right quadrant denote a bias for coordinating on B’s color, locations in the lower left quadrant a bias in coordinating on A’s preferred color. Monkeys F (orange) and C (green) in late confederate training sessions showed balanced selfish/benevolent coordination throughout each session, as well as 4 late macaque pairs that coordinated on a fixed side (center area). Macaque pairs coordinating on a fixed color (located around [0,0] or [1,1]) also did not shift their behavior in the course of a session.

“To assess if the balance between selfish vs benevolent coordination changed in the course of each session, we analyzed the proportion of coordinating on F’s preferred color among all coordinated trials (note that this is complementary to the proportion of coordinating on C’s color) in first 100 vs last 200 trials (Fisher’s exact test with Bonferroni correction for the number of tested sessions). In several sessions (FC-1, FC-2, FC-3, FC-6) one of the two agents became more selfishly insistent on his preferred color towards the end of the session – while the other accommodated. Interestingly, the roles reversed in different sessions: in FC-1 C is the leader, while in FC-3 and FC-6 F is the leader. We conclude that during the dynamic coordination, there is a tendency for this pair to gradually adopt more consistent leader-follower roles, potentially because one monkey accommodates and follows more easily in a more sated state.”

Furthermore, we added the number of solo trials monkeys performed after each dyadic session, to show that the end of dyadic sessions did not correspond to fully waning motivation (lines 864-866):

“After the dyadic task we offered monkeys the opportunity to obtain additional liquid in the solo version of the task for as long as they continued to work (mean 169 trials, median 73, range 0 to 1163), essentially making participation in the dyadic trials optional.”

To sum up, while it is indisputable that the nature of the reward and the underlying motivations differed between the two species, there is very little indication that variations in satiation level in the course of each session played a substantial role in shaping macaque strategies – further highlighted in lines 682-691:

“Inherently, the nature of reward differed between the species: humans received monetary compensation while macaques worked for a liquid reward. Note, however, that in each session macaques were offered an opportunity to work and obtain more liquid in a solo regime after the dyadic experiment (Materials and methods), so the participation in a dyadic interaction was not forced, and the end of the dyadic sessions was not associated with a fully waning motivation. We also did not observe that variations in satiation level in the course of each session played a substantial role in shaping macaque strategies. Nevertheless, it is important to acknowledge differences between primary and secondary rewards, and potential divergence in the underlying motivations and learning, beyond the species differences in socialization and cognitive capacities considered above.”

5. Can the authors more precisely consider the role of action visibility? Specifically, whether visibility influences social cognition or aides sensorimotor coordination.

This point is related to the question raised by Reviewer 2, Recommendations to the authors Comment 3.

We thank the Reviewer for highlighting this aspect. The answer is both. Action visibility is certainly influencing the sensorimotor coordination, as demonstrated by the differences in action times between opaque and transparent conditions (Figures 4C and S4C), convergence to a faster agent’s preference during static coordination (Table S2, Supplementary file 2), a tight relationship between action visibility and choice behavior (Figures 6 and S7), and inter-agent action time differences in coordinated vs non-coordinated trials (Table S1, Supplementary file 1, see also point 1). Furthermore, anecdotally human confederates reported that macaques paid close attention to their actions and often did not even start the trial unless they saw a human doing so. At the same time, we firmly believe (but can only speculate, based on our own impressions and the informal feedback from human subjects) that also the social cognition “per se” was influenced by the direct face-to-face context and by touching the “same” targets. We expanded discussion on this in lines 582-589:

“In our study, action visibility facilitated sensorimotor coordination, as demonstrated by action time differences in transparent vs. opaque conditions, convergence to a faster agent’s preference during static coordination, a tight relationship between action visibility and choice behavior in turn-taking macaques, and inter-agent action time differences in coordinated vs. non-coordinated trials. Seeing actual movements rather than relying on abstract representations of others such as cursor motions might be especially crucial in nonhuman experiments. But beyond immediate sensorimotor aspects, acting face-to-face on the same targets in close proximity likely influenced social cognition on a more general level.”

6. Finally, can the authors provide further discussion of the effects of learning and socialization on this task across species?

This point is related to the question raised by Reviewer 3. We readily acknowledge the differences between primary vs secondary rewards (please see also point 4 above) and different underlying motivations and expectations of the two species. We see within-session learning effects in both species, but only in monkeys we could study learning across multiple sessions, with the same and different partners. We extended the acknowledgment of these differences in lines 688-696:

“Nevertheless, it is important to acknowledge differences between primary and secondary rewards, and potential divergence in the underlying motivations and learning, beyond the species differences in socialization and cognitive capacities considered above.

Finally, it would be very interesting to relate diverse in-game behavioral strategies observed in the human pairs to individual differences in cognitive and socio-economic personality traits (Rollwage et al., 2020; Zhao and Smillie, 2015), and link them to stable behavioral phenotypes (Peysakhovich et al., 2014; Poncela-Casasnovas et al., 2016). This however will require a much larger sample and personality-oriented study design.”

Reviewer #1 (Recommendations for the authors):

1. It would be helpful to know more about how the goal of maximizing reward might impact the behaviors described here. To start, how does turn-taking compare with the simpler strategies used by the monkeys in terms of reward received? Figure 2C seems to show that the average reward earned by two subjects was actually quite similar between monkeys and humans. The highest-earning human pairs (presumably those who were turn-taking) might have earned slightly more than the monkeys, but there doesn't appear to be a big advantage to turn-taking, in terms of simple reward maximization. Is this true, and if so, is it possible to interpret their behavior as arising from other advantages (e.g., placing value on a partner's reward specifically vs. perceived equity)? What about rewards received by each individual and not the average across pairs? If an agent only values rewards received by themself, is there a more or less optimal approach to this task?

Please see the response to the point 2 of essential revisions.

In brief, the revised Figure 2C and associated analysis indeed show that the average joint reward was similar between monkeys and humans, and the advantage of the turn-taking (or a side strategy resulting in “automatic turn-taking” adopted by 4 monkey pairs) is not in maximizing the joint reward per pair, but in simultaneously maximizing both individual rewards. We surmise that human turn-takers – but not side-coordinating macaques – were indeed at least in part motivated by equity considerations.

The individual rewards are shown in Figure 2—figure supplement 2B,G. Indeed, if the agent only values own reward, the optimal approach is to force coordination on his/her preferred color, as long as the partner plays along. Clear examples of such behavior are human pairs 10, 18, 16 and macaque pairs CL, TF and TC (individual rewards ~4 and ~3). From a purely economic value perspective of a selfish agent, the turn-taking only makes sense to ensure coordination in a long term (to avoid a scenario where previously accommodating partner is fed up and starts refusing to coordinate at all on the selfish agent’s color).

2. Related to the above, it is notable that rewards given to the monkeys and humans differed qualitatively, as did their motivation to pursue them. The humans were receiving small amounts of money in a fairly abstract arrangement in which money was obtained at the end of the session, whereas the monkeys were thirsty and received fluid rewards continuously depending on trial outcomes. To what extent is the motivation to be 'selfish' versus cooperative driven by the motivation to obtain each outcome? For instance, if monkeys were not water deprived, would they be more likely to cooperate rather than compete? Is there any evidence that motivation to get the juice reward changed toward the end of the session?

Please see the response to the point 4 of essential revisions.

3. Are there action time plots similar to those shown for the monkeys for the human pairs? Although Figure 6B is compelling, if I understand it correctly, it just shows that the humans' choices can't be predicted based on who responded first. But it's hard to understand, then, how this type of coordination would emerge if one participant didn't watch and follow the other. The results in 6B could occur if the person who responds faster also drives the shifts to the other's target from time to time, as suggested in Figure 7. Therefore, is it possible that there is a leader and follower in the human pairs (perhaps these roles switch off), but the leader just doesn't choose their own color reliably?

Please see the response to the point 1 of essential revisions.

Since the point 1 of essential revisions also deals with a different emphasis on comparing turn-takers vs non turn-takers, raised by the Reviewer 2, we briefly summarize the specific answer here. The Reviewer is absolutely correct – indeed, there were temporal leaders and followers in the human pairs, and indeed, the faster players equally often selected their own color and the color of the partner, to equalize the rewards, as suggested by Figure 7 and Figure 7—figure supplement 1. We now emphasize this point more clearly in lines 422-426:

“Hence, similarly to macaques, seeing the faster agent’s choice can help the slower agent to coordinate. But unlike the confederate-trained macaque pair, dynamic human coordination was not driven by the faster agents striving to select only the own preferred color (except human pair 12 shown in Figure 6B): for instance, in the turn-taking pairs 3 and 17, the faster agent B led both to own and A’s preferred colors, while A followed (Table S1, Supplementary file 1).”

Figure 6—figure supplement 1 and Table S4 (Supplementary file 4) show correlation coefficients between the visibility of other’s actions and selecting other’s color for all human dyads. The pair 12 shown in Figure 6B shows the highest correlation among the human dyads. In this pair, firstly player A led (equally often to own and to B’s color) and B followed, and then B led (more to own color) and A followed. Among all human pairs, only in this pair there was a reliable difference between AT A – AT B for coordination on A’s color vs coordination on B’s color, but even here the separation between the AT difference histograms was not as clear as in the macaque pair FC (Author response image 5).

**Author response image 5. sa2fig5:** Action visibility and action time differences. (A) This panel repeats Figure 6B for convenience; it shows the correlation between probability of seeing the other move first (in yellow) and following to the other’s preferred target for human pair 12 (top/red for agent A; bottom/blue for agent B). (B) This panel shows the “fraction of choosing own” graph for the same pair (agent A’s in red, agent B’s in blue). The pair starts with a balanced fluctuations selecting both A’s and B’s colors and ends with mostly convergence on B’s color. (C) This panel shows the histogram of the action times for the different choice combinations (red: both choose A’s preferred color; blue: both choose B’s preferred color). The mean action time difference between agent A and agent B differed significantly when the pair coordinated on A’s color compared to B’s color (t-test: coordination on red (M: -88.15, SD: 128.96, N: 96) vs. blue (M: 12.44, SD: 109.49, N: 180), t(168.8392): -6.4951, p < 0.00001). In other words, when jointly selecting red, A was on average 88 ms faster than B; when jointly selecting blue, A was on average 12 ms slower than B..

Reviewer #2 (Recommendations for the authors):This manuscript is nicely organized and very thorough. However, it could be improved even further if some points can be discussed in more detail or with additional data analyses.1. Related to Figure 2D: the authors mentioned that some human pairs (about half) were not employing the turn-taking strategy. From this figure, these pairs were actually not a small number (9 out of 19). The authors should discuss what might be the reason behind this. Is it possible that these subjects were not very socially engaged (i.e., not paying attention to others) or were less flexible when choosing different options (even in non-social situations)? It will be better if the authors could perform additional data analyses to characterize the differences between the two groups of humans (social attentional differences, reaction time differences, etc) to examine why they were different from the turn-taking pairs and more similar to the monkey pairs.

Please see the response to the point 1 of essential revisions.

2. Related to Figure 5: It is still a little puzzling that monkeys' behaviors can be improved after being trained with humans. What was the reason for this change? Did the human subjects provide extra social cues to the monkeys in order to encourage them to pay more attention, or did the monkeys learn turn-taking simply from the diverse choice patterns of the human subjects? The authors need to have further discussion about this point.

Please see the response to the point 3 of essential revisions.

3. It is not very clear whether the action visibility is influencing social cognition or simply aiding sensorimotor coordination. The authors should discuss this further in Discussion.

Please see the response to the point 5 of essential revisions.